# Deciphering the complex role of thrombospondin-1 in glioblastoma development

Thomas Daubon[1,2,3,4], Céline Léon[1,2], Kim Clarke[5], Laetitia Andrique[1,2], Laura Salabert[1,2], Elodie Darbo[6], Raphael Pineau[7], Sylvaine Guérit[1,2], Marlène Maitre[8], Stéphane Dedieu[9], Albin Jeanne[9,10], Sabine Bailly[11], Jean-Jacques Feige[11], Hrvoje Miletic[3,12], Marco Rossi [13], Lorenzo Bello[13], Francesco Falciani[5], Rolf Bjerkvig[3,4,14] & Andréas Bikfalvi[1,2]

We undertook a systematic study focused on the matricellular protein Thrombospondin-1 (THBS1) to uncover molecular mechanisms underlying the role of THBS1 in glioblastoma (GBM) development. THBS1 was found to be increased with glioma grades. Mechanistically, we show that the TGFβ canonical pathway transcriptionally regulates THBS1, through SMAD3 binding to the THBS1 gene promoter. THBS1 silencing inhibits tumour cell invasion and growth, alone and in combination with anti-angiogenic therapy. Specific inhibition of the THBS1/CD47 interaction using an antagonist peptide decreases cell invasion. This is confirmed by CD47 knock-down experiments. RNA sequencing of patient-derived xenograft tissue from laser capture micro-dissected peripheral and central tumour areas demonstrates that THBS1 is one of the gene with the highest connectivity at the tumour borders. All in all, these data show that TGFβ1 induces THBS1 expression via Smad3 which contributes to the invasive behaviour during GBM expansion. Furthermore, tumour cell-bound CD47 is implicated in this process.

[1] INSERM U1029, Institut Nationale de la Santé et de la Recherche Médicale, 33615 Pessac, France. [2] University Bordeaux, 33615 Pessac, France. [3] KG Jebsen Brain Tumor Research Center, University of Bergen, 5020 Bergen, Norway. [4] Norlux Beuro-Oncology, Department of Biomedicine, University of Bergen, 5020 Bergen, Norway. [5] Computational Biology Facility, University of Liverpool, Liverpool L69 7ZB, UK. [6] UMR1218 ACTION, Bioinformatic Center CBiB, University of Bordeaux, 33076 Bordeaux, France. [7] Animal Facility, University Bordeaux, 33615 Pessac, France. [8] INSERM U1215, Neurocenter Magendie, Pathophysiology of Addiction Group, 33076 Bordeaux, France. [9] CNRS UMR 7369, MEDyC, 51687 Reims, France. [10] SATT Nord, 59800 Lille, France. [11] INSERM U1036, Grenoble 38000, France. [12] Department of Pathology, Haukeland University Hospital, 5020 Bergen, Norway. [13] Neurosurgical Oncology Unit, Department of Oncology and Hemato-Oncology, Humanitas Research Hospital, Universita Degli Studi di Milano, 20089 Rozzano, Milan, Italy. [14] Oncology Department, Luxembourg Institute of Health, 84, Val Fleuri 1526, Luxembourg. These authors contributed equally: Rolf Bjerkvig, Andréas Bikfalvi. Correspondence and requests for materials should be addressed to T.D. (email: thomas.daubon@u-bordeaux.fr) or to A.B. (email: andreas.bikfalvi@u-bordeaux.fr)

Gliomas are classified by the WHO in four grades where glioblastoma (GBM) represents the most aggressive form[1]. GBMs are diagnosed based on their neuropathological features including mitotic activity, diffuse invasion, and extensive angiogenesis and necrosis, the latter associated with blood–brain barrier disruption[2]. Cell invasion represents a key challenge for effective drug delivery and may be induced by anti-VEGF therapy[3].

The extracellular matrix (ECM), composed of fibrous proteins and glycoproteins, was first acknowledged for its scaffolding role, but is now recognised to have a role in numerous physiological and pathological processes[4]. This includes tumour development and metastasis[5], where there is a striking difference between the ECM of normal tissue as compared with tumours[6]. The thrombospondins, a family of five members (THBS1-5), are important components of the ECM[7]. THBS1 was first discovered in platelets but has now been shown to have an important role in cancer development[8,9]. Besides having a direct role in regulating tumour cell behaviour, THBS1 also exhibits functions in the tumour vasculature[10]. Numerous studies on several cancer types including GBM[11] indicate that THBS1 can modulate immune responses as well as GBM vascularisation[12]. However, the precise contributions of THBS1 in GBM development as well as its regulation have not yet been fully determined.

Crosstalk between tumour and endothelial cells are driven by several factors including VEGF and TGFβ1, the latter being known to have a central role in GBM development[13]. We have previously shown that THBS1 is expressed in tumour blood vessels and in specific patient-derived xenograft (PDX) models[14]. It has been proposed that THBS1 activates TGFβ1 via its type 1 domain by mobilising its active form from the Latent Activating Protein (LAP)[15]. However, this may apply only to some but not all tumour types and it is still a matter of debate[16]. It has also not been established how TGFβ1 itself is able to regulate THBS1 expression[17]. THBS1 interacts with many effector proteins, including α6β1 or α4β1 integrins, as well as with cell-surface receptors such as CD36 and CD47[7]. THBS1/CD47 interactions have been reported to be important in vascularisation and tumour progression[18], but have not been associated with GBM growth.

In this study, global expression analysis revealed THBS1 to be upregulated in high-grade gliomas and to be associated with a poor prognosis. Furthermore, we found that TGFβ1 activation was not regulated by THBS1 in GBM, but on the contrary, TGFβ1 induced THBS1 expression through direct transcriptional activation via SMAD3. Our data show that THBS1 is not only involved in the regulation of angiogenesis in GBM, but also impacts the invasive behaviour of glioma cells and that THBS1/CD47 interactions contributes to this process. Finally, we performed gene expression analysis by RNA-sequencing after microdissection of central and peripheral tumour areas in a human PDX model[3,19]. We show striking differences between both areas in the tumour and stromal cell compartments. In this analysis, THBS1 was the gene with the highest connectivity in the peripheral tumour areas.

## Results

### THBS1 is differently expressed between the glioma grades.
THBS1 has a role in tumour invasion in vivo in prostate cancer[9] and medulloblastoma[20], but its putative role in GBM invasion has not been explored so far. We have previously shown that THBS1 is expressed in tumour blood vessels and in specific PDX models[14]. According to The Cancer Genome Atlas (TCGA), THBS1 expression is found to be increased in GBMs when compared to grade II and III tumours (Supplementary Fig. 1A), and linked to

patient survival (Supplementary Fig. 1B). TGFβ1 expression was only slightly upregulated in high-grade gliomas (Supplementary Fig. 1A) and linked to survival (Supplementary Fig. 1B).

THBS1 was assessed for expression in patient samples of glioma grade II, III and IV by immunohistochemistry (IHC) (Fig. 1a). THBS1 was expressed at higher levels in GBM when compared to glioma grades II, III, or normal brains (Fig. 1b). When patient GBM tumour samples from multiple areas were analysed by IHC, a specific pattern of THBS1 staining was seen in the central tumour area and at the location of invasive cells (Supplementary Fig. 2A, B). THBS1 was detected in both, tumour vessels and tumour cells, the latter being enhanced in the peripheral invasive areas (margin) (Supplementary Fig. 2A, B). Significant colocalisation between THBS1 and the hypoxia marker CAIX (Carbonic Anhydrase IX) was detected in the patient GBM core, but only single positive cells (CAIX or THBS1) were detected in the invasive area (Supplementary Fig. 2C).

The P3 human xenograft model has previously been characterised in several studies[3,21,22]. P3 give rise to tumours that show a close resemblance to human GBMs, exhibiting a necrotic core surrounded by pseudo-palisading cells as well as angiogenic and invasive areas[14]. IHC of P3 xenografts demonstrated higher THBS1 deposits within invasive areas (Fig. 1c, right panel) (Nestin staining of the invasive area of the corpus callosum is shown in Fig. 1c, left panel).

### TGFβ1 controls THBS1 expression via SMAD3 binding.
We have previously shown that inactivation of the stress response protein IRE1 leads to a change from an angiogenic to an invasive phenotype in U87 tumours[23]. In this case, THBS1 was found to be the most upregulated gene at both the mRNA and protein level[23]. Western blot analyses confirmed THBS1 upregulation in U87 IRE1dn cells, in which Xbp1 is not spliced, as compared with U87 control cells (Supplementary Fig. 3A). Furthermore, TGFβ1 activity was found increased in IRE1dn vs control cells as demonstrated by measurements of active TGFβ1 (Supplementary Fig. 3B) and nuclear localisation of P-SMAD2 (Supplementary Fig. 3C). This suggested a link between TGFβ activity and THBS1 expression in our glioma model. To investigate the putative involvement of THBS1 in direct TGFβ activation, U87wt cells transfected with a SMAD luciferase reporter were incubated with recombinant THBS1 to monitor TGFβ activation (Supplementary Fig. 3D). Alternatively, U87wt cells stimulated with recombinant THBS1 were probed by western blot for TGFβ effectors (P-SMAD2 and FIBRONECTIN), and for PHOSPHO-AKT as a control of THBS1 receptor activation (Supplementary Fig. 3E, left panel). Under these conditions, THBS1 did not modulate the TGFβ pathway in the presence or absence of LSKL peptide, a putative inhibitor of THBS1/latent-activating protein interaction[24] (Supplementary Fig. 3D–E). However, P-AKT increase was seen after short-term THBS1 stimulation (Supplementary Fig. 3E, right panel). Furthermore, we demonstrated that knockdown of THBS1 in U87 IRE1dn cells, which endogenously express high levels of THBS1[23] (Supplementary Fig. 3A), had no effect on both the concentration of active TGFβ in the medium (200–300 pg/10^6 cells, Supplementary Fig. 3F) or TGFβ activity measured by nuclear P-SMAD2/3 accumulation (Supplementary Fig. 3G). These results indicate that THBS1 has no direct role in TGFβ activation in the glioma model.

The patient-derived cell line P3 and U87 cells similarly responded to short-term TGFβ1 induction by increasing SMAD2 and SMAD3 phosphorylation (Supplementary Fig. 4A). After longer TGFβ1 stimulation, we found an increase in THBS1 expression in both P3 and U87 cells (respectively, $2.6 \pm 0.007$- and $6.94 \pm 0.17$-fold increase of THBS1 mRNA, $1.95 \pm 0.16$- and

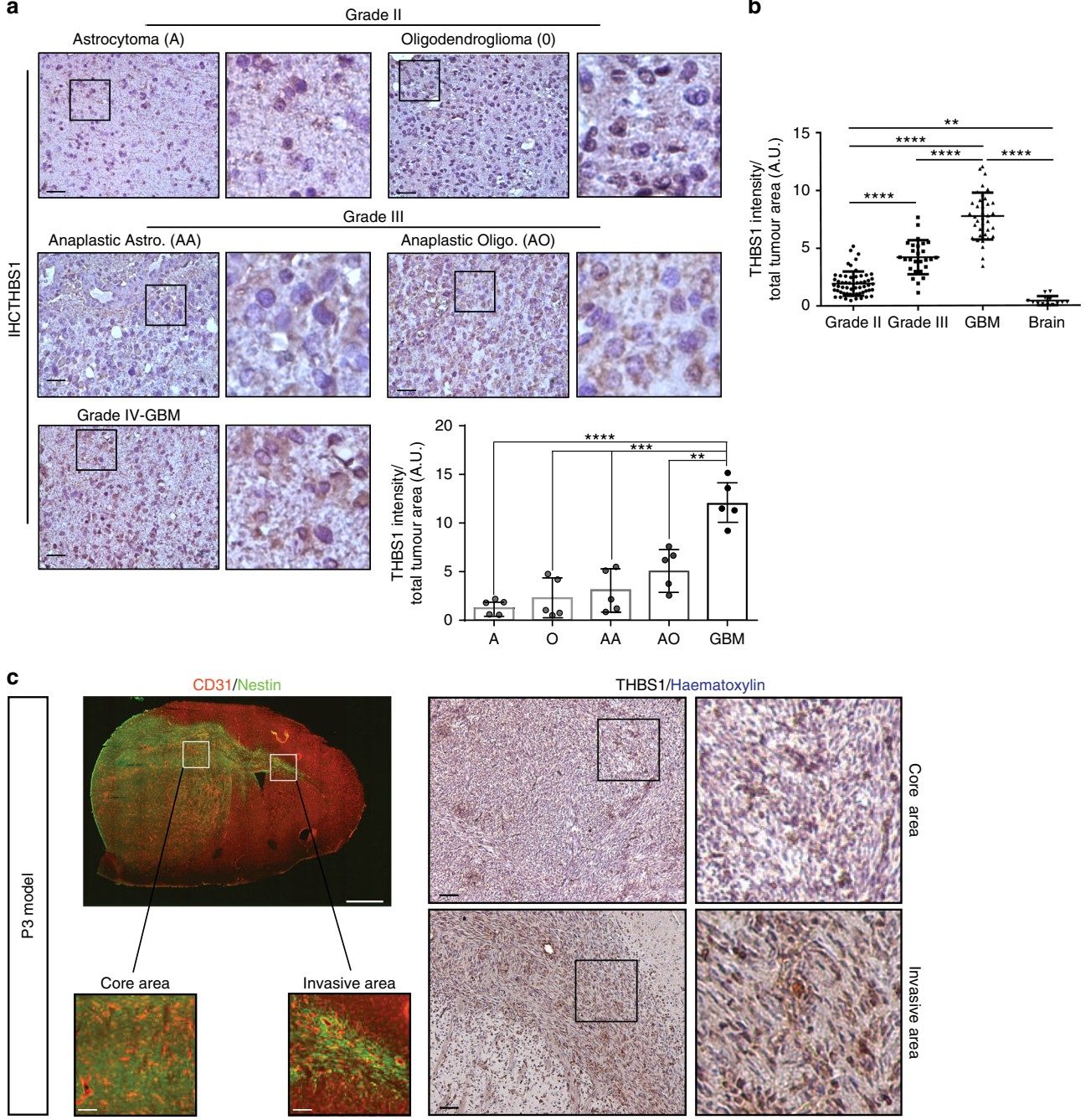

**Fig. 1** THBS1 is a marker of high-grade glioma. **a** Immunohistochemistry (IHC) of THBS1 in patient samples from grade II astrocytomas and oligodendrogliomas (upper panels), grade III anaplastic astrocytomas and anaplastic oligodendrogliomas (middle panels) and grade IV glioblastomas (lower panels). Magnification of tumour areas is shown in each panel on the right hand side. Scale: 50 μm. Quantification of THBS1 staining by IHC profiler from Fiji software. A, astrocytomas (n = 5); O, oligodendrogliomas (n = 5); AA, anaplastic astrocytomas (n = 5); AO, anaplastic oligodendrogliomas (n = 5); and GBM, glioblastomas (n = 5). The graph represents the results as means ± s.d. **P < 0.01; ***P < 0.001; ****P < 0.0001; ns not significant (ANOVA). **b** Quantification of THBS1 staining by using tissue microarrays. 121 patients were analysed using TMA (Biomax), and IHC with anti-THBS1 antibodies. 57 were of grade II, 27 of grade III and 37 of GBM–grade IV. The graph represents the results as means ± s.d. **P < 0.01; ****P < 0.0001 (ANOVA). **c** Left panels: immunofluorescence stainings of CD31 (red) and Nestin (green) of a P3 tumour section. Images below represent core and invasive areas. Scales: 500 μm (upper panel) and 20 μm (lower panels). Right panels: IHC for THBS1 in samples representing core and invasive areas in P3 tumours, counterstained with haematoxylin. THBS1 is expressed in invasive areas of P3 tumours. Scale: 50 μm

1.67 ± 0.07-fold increase of THBS1 protein expression) (Fig. 2a). To corroborate our observations, we stimulated two other patient-derived cell lines (BL9 and BL13) with recombinant TGFβ1, and observed a time-dependent increase in THBS1 expression (Supplementary Fig. 4B). THBS1 expression was only increased after TGFβ1 stimulation, not after TGFβ2, EGF, PDGF-BB, or IL-1β stimulation (Supplementary Fig. 4C). Immunostaining revealed that THBS1 appeared as small aligned dots along cell protrusions that were further increased under TGFβ1 stimulation (Supplementary Fig. 4D). THBS1 expression and secretion were measured by ELISA in both cell extracts and in the extracellular environment (free-THBS1) (Fig. 2b). Induction by TGFβ1 was

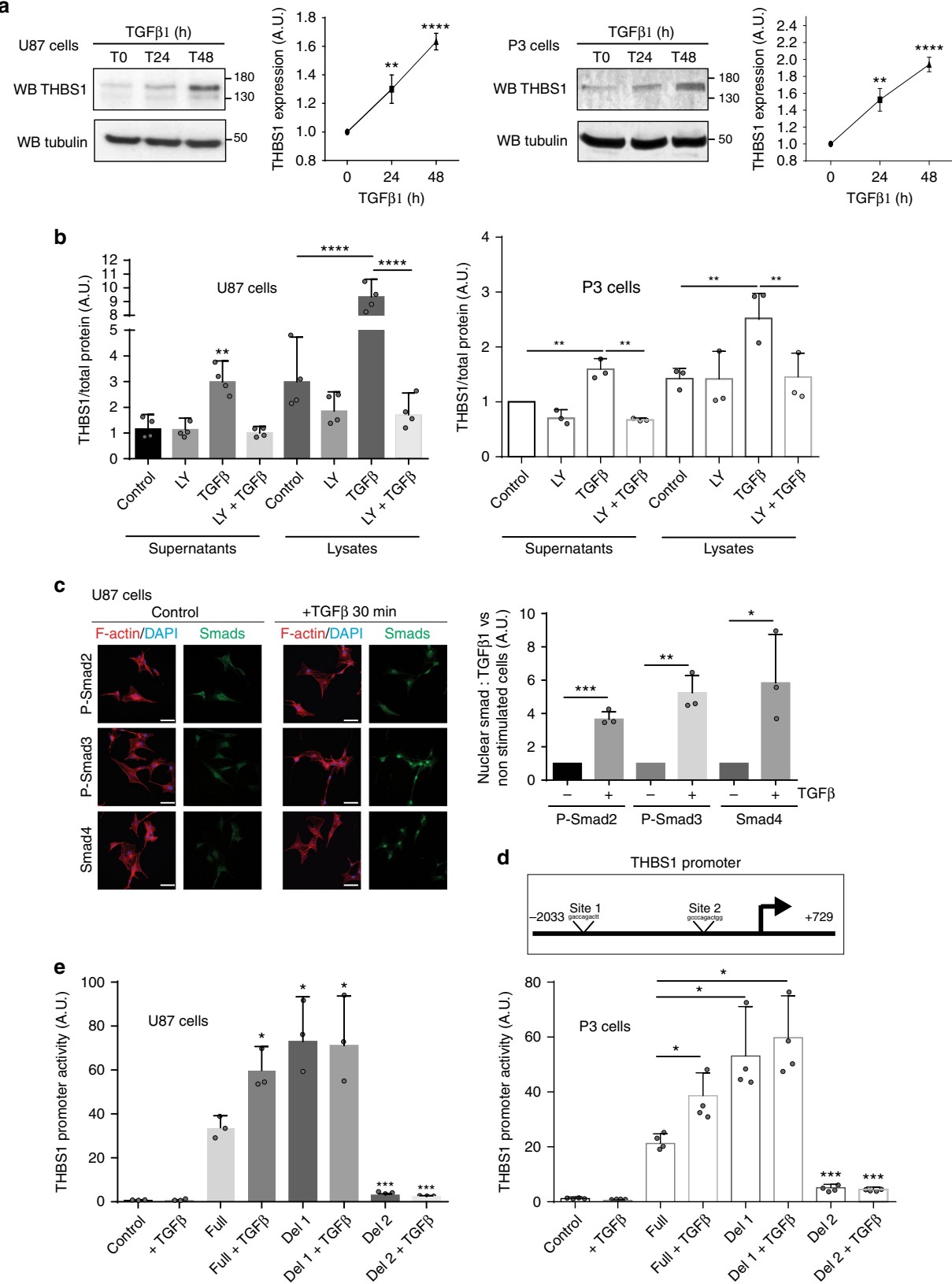

completely blocked by the specific TGFβ receptor inhibitor, LY2157299, in cell lysates, but also in supernatants (Fig. 2b). To analyse canonical downstream signalling of TGFβ1, immunostainings for P-SMAD2, P-SMAD3 and SMAD4 were performed. All SMAD proteins were found to be located in the nuclei of U87

cells after TGFβ1 stimulation (Fig. 2c). Only binding sites for Smad3 were found in the human THBS1 promoter. These are located at −1125/−1115 for site 1 and at −110/−100 for site 2 (Fig. 2d). Importantly, SMAD3 silencing by siRNAs decreased TGFβ1-induced THBS1 expression with a stronger effect for

**Fig. 2** THBS1 expression is regulated by TGFβ1 via SMAD3 promoter binding. **a** Analysis of THBS1 expression in protein extracts from non-treated or TGFβ1-treated U87 (left) and P3 (right) cells (24 or 48 h treatment). TGFβ1 was used at a concentration of 5 ng/ml. The graphs represent quantifications of THBS1 signal normalised to Tubulin ($n = 3$). Densitometry analysis (right panels) is represented as fold induction compared to control. Student's $t$-test $P$-value: **$P < 0.01$; ****$P < 0.0001$. **b** ELISA experiment performed on U87 (left) and P3 (right) cell supernatants or lysates, treated with 5 ng/ml of TGFβ1, 10 µM of TGFβR inhibitor LY2157299 or combination treatment for 48 h. The graph represents a mean of four (U87) or three (P3) independent experiments as fold induction to non-treated cells. **c** Representative immunofluorescence images of starved U87 control or TGFβ1-treated cells (30 min TGFβ1 treatment with a concentration of 5 ng/ml) showing nuclear translocation of P-SMAD2, P-SMAD3 or SMAD4. Staining: Smads (green), Phalloidin for F-Actin (red) and DAPI for nuclei (blue). Scale bars: 10 µm. The graph on the right represents a mean of three independent experiments (100 cells analysed in each), as fold induction vs non-treated cells. **d** Schematic representation of SMAD3-binding sites on *THBS1* promoter. Binding site 1 is located at −1125/−1115 and binding site 2 at −110/−100 on the *THBS1* promoter. **e** Luciferase promoter activity was measured by transfecting a PGL3 vector containing the full-length THBS1 promoter (Full) or inserts deleted either in binding site 1 (Del1) or binding site 2 (Del2). A GFP plasmid was used as a control. U87 (left) and P3 (right) cells were starved for 24 h and treated with 5 ng/ml of recombinant TGFβ1. The results are represented of three (U87) or four (P3) independent experiments. All graphs are represented as means ± s.d. *$P < 0.05$; **$P < 0.01$; ***$P < 0.001$; ns, not significant (ANOVA)

SMAD3-2 siRNA (Supplementary Fig. 5A, B). To acquire a deeper understanding of THBS1 transcriptional regulation, we mutated the putative SMAD3-binding sites on the *THBS1* promoter and transduced P3 and U87 cells with these constructs. Luciferase experiments showed that *THBS1* promoter activity is inhibited when the second SMAD3-binding site was mutated, while increased when the first binding site was mutated (Fig. 2e). These results indicate a direct role for canonical TGFβ1 signalling in controlling *THBS1* transcriptional activity via SMAD3.

**THBS1 impacts on tumour expansion and invasion.** First, a THBS1 shRNA strategy was employed to inhibit THBS1 expression in U87wt cells. THBS1 expression was strongly reduced with both shRNAs (Supplementary Fig. 6A). Furthermore, invasion was significantly decreased in THBS1-knockdown U87 cells compared to control cells (Supplementary Fig. 6B). In P3 spheroids, THBS1-knockdown (Supplementary Fig. 7A) led to a significant reduction in cell invasion when tested in a collagen type I invasion assay (Fig. 3a). Knockdown THBS1 spheroids were then orthotopically implanted in mice, and tumours were then analysed by IHC (Supplementary Fig. 7B). Differences in vessel type were found in the THBS1-depleted tumours when compared to control, with a decrease in small vessel number (<10 µm length) and an increase in medium size vessels (between 10 and 20 µm length) (Fig. 3b). Contra-lateral invasion was significantly decreased by more than 49% for P3 THBS1-1 shRNA tumours and 76% for P3 THBS1-2 shRNA tumours in comparison with shRNA control tumours (Fig. 3b). Survival was significantly increased for mice implanted with P3 tumours transduced with either one of the THBS1 shRNA constructs (Fig. 3c). To reinforce these results, we performed a gain-of-function experiment by expressing THBS1 in P3 cells using a lentiviral construct. Overexpression was ascertained by western-blot and immunofluorescence (Fig. 3d). In this experiment, invasion of THBS1-overexpressing cells was increased (Fig. 3d).

**Hypoxia increased THBS1 expression through TGFβ1 activation.** Anti-angiogenic therapy using anti-VEGF antibodies (bevacizumab) is commonly used for the treatment of recurrent GBMs[25]. Tumour hypoxia and a subsequent increase in HIF1α expression are consequences of bevacizumab treatment in vivo, which may lead to increased local invasion[26]. Since it has been recently shown that bevacizumab treatment leads to an increase in TGFβ1 mRNA expression in GBM[27], we wanted to further assess the function of THBS1 following bevacizumab treatment. Bevacizumab-treated U87 tumours exhibited some invasive borders characterised by THBS1 deposits in the vicinity of invasive strands (Supplementary Fig. 8A). In bevacizumab-untreated P3

tumours, THBS1 deposits were already observed in peripheral areas but not in the tumour core (Fig. 4a). This was further confirmed by laser-capture microdissection and qPCR analysis where THBS1 mRNA was found significantly increased in the invasive area when compared with the tumour centre (Fig. 4b). In bevacizumab-treated P3 tumours, an increase in THBS1 expression was seen in the tumour core by western blot (Fig. 4a). Moreover, under in vitro hypoxic conditions, P3 and U87 cells increased THBS1 expression, as demonstrated by western blot (Fig. 4c and Supplementary Fig. 8B). Hypoxia activated the SMAD pathway as revealed by luciferase reporter experiments (Fig. 4d and Supplementary Fig. 8C), and we showed by using ELISA that THBS1 expression and secretion was inhibiting using LY2157299 (Fig. 4e and Supplementary Fig. 8D). In the highly angiogenic U87 glioma model treated with bevacizumab, silencing of THBS1 by shRNA led to a decrease in tumour expansion and invasion (Supplementary Fig. 8E). Survival was significantly increased when both THBS1 shRNA-transduced U87 and P3 tumours were treated with bevacizumab (Fig. 4f and Supplementary Fig. 8F).

**CD47/THBS1 interaction is involved in GBM expansion.** It has recently been shown that TAX2, a 12 amino-acid peptide stretch of CD47, specifically inhibits the THBS1/CD47 interaction, causing necrosis in various tumour models[18,28]. In particular, TAX2 inhibits THBS1/CD47 interactions at the surface of endothelial cells. P3 cells highly express CD47 (Supplementary Data 1) while exhibiting low CD36 in vivo expression (Supplementary Data 1).

Since hypoxia induces THBS1 expression (Fig. 4c), we treated P3 spheroids with TAX2 or control peptide under hypoxic conditions (Fig. 5a). In this case, TAX2 significantly inhibited hypoxia-induced in vitro cell invasion (Fig. 5a). We next investigated the effect of the various treatments on tumour development in vivo. Blood vessel density was modified in all treatment groups with a strong effect of the TAX2 + bevacizumab combination on small vessels (Fig. 5b). Combination treatment inhibited contra-lateral and single-cell invasion in the P3 intracranial tumour model in comparison to bevacizumab treatment alone (Fig. 5c). Survival was increased when animals were treated with combinatory treatment (Fig. 5d).

**CD47 is a key player of GBM invasion.** We have furthermore reinforced our results by silencing CD47 in both P3 and U87 cells. We have used single-cell track migration as well as spheroid invasion assay. The spheroid invasion assay in collagen I with CD47-silenced P3 cells showed significant reduction of invasion (Fig. 6a, b). The single-cell migration track assay with CD47-silenced U87 cells demonstrated significant inhibition of cell

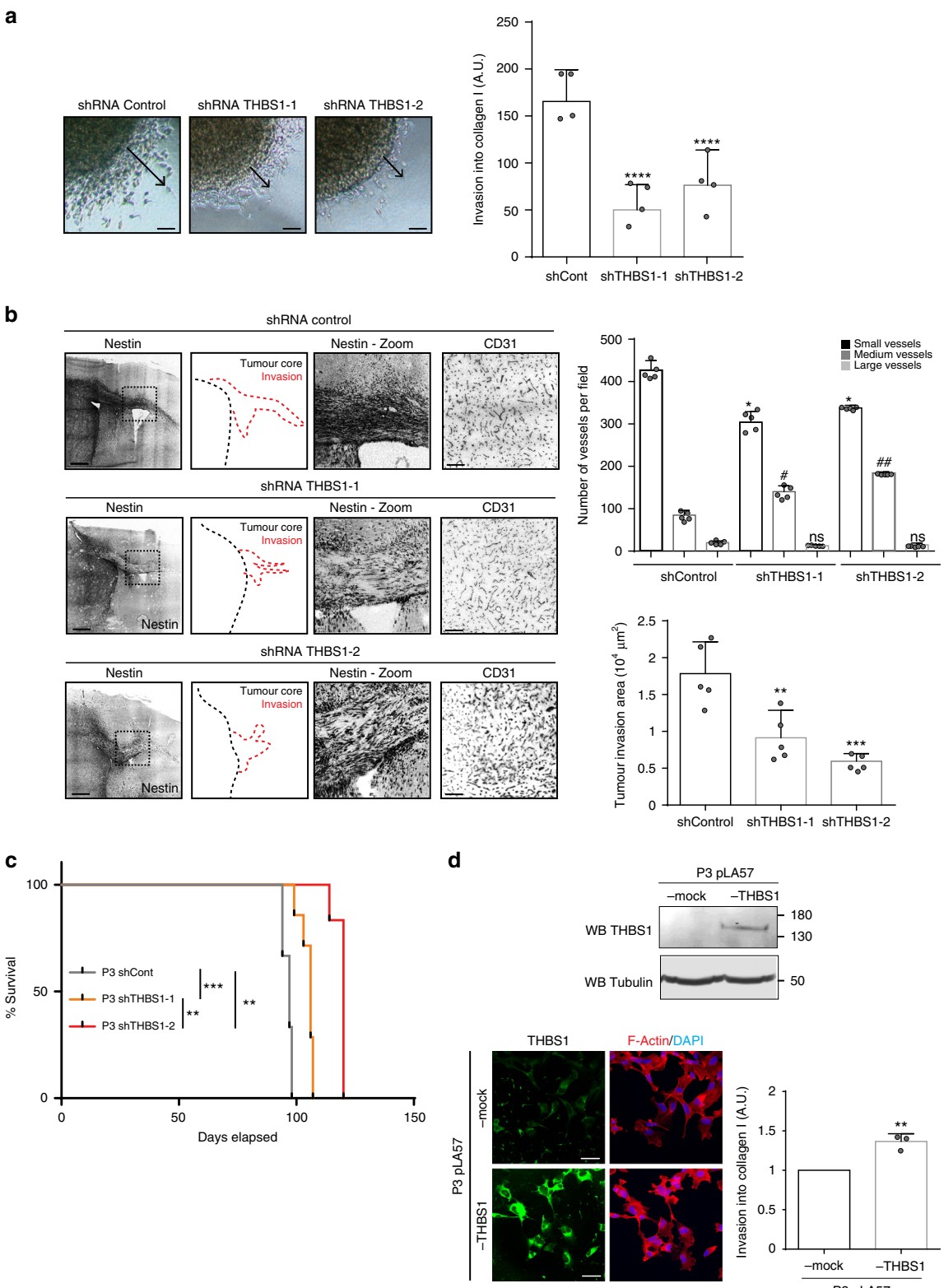

migration in comparison to U87 control cells (Supplementary Fig. 9A, B). Next, we have conducted in vitro invasion experiments in P3 shRNA CD47 or control cells stimulated or not with purified full-length THBS1. The results show that knockdown of CD47 significantly impairs THBS1-induced tumour cell invasion (Fig. 6c). Furthermore, survival studies in mice implanted with P3 CD47 knockdown or control cells showed significant increase in survival times in the CD47 knockdown group (Fig. 6d).

**THBS1 is the gene with highest connectivity**. To reinforce our results at a global scale, we performed transcriptomics analysis via RNA-sequencing of P3 tumours. GFP-labelled tumour cells from both invasive and angiogenic areas of P3 were visualised and then laser-microdissected prior to RNA isolation and sequencing (Fig. 1c). Gene expression profiling and data analysis were performed with respect to human and mouse RNA independently using Xenome (Supplementary Data 1). Principle component

**Fig. 3** THBS1 controls invasion and growth of P3 tumour. **a** P3 cells were transduced with shRNA control or shRNA THBS1 (−1 and −2). P3 spheroid invasion was measured in collagen I gels after 24 h. Arrows represent the migration distance from the spheroid core. Scale: 50 µm. The graph represents the invasion results as means ± s.d. of four independent experiments each done in 6–8 replicates for each condition. ****$P < 0.0001$ (ANOVA). **b** Control shRNA or THBS1 shRNA-transduced P3 cells were xenotransplanted into Ragγ2 C$^{-/-}$ mice. Representative images of tumours with invasive areas are shown by Nestin staining (grey) with schematic representations (invasion in red dashed lines and tumour mass in black dashed lines). Scale: 200 µm. The graphs on the right represent the number of small (<10 µm), medium (between 10 and 20 µm), large blood vessels (>20 µm) (upper panel) and the invasion area (lower panel) of control shRNA and THBS1-1/-2 shRNA-transduced P3 tumours (average of 5 tumours with 8 sections/tumour analysed; * means comparison of small vessels; # means comparison of medium size vessels). The graphs represent results as means ± s.d ($n = 5$ tumours analysed per group). *$P < 0.05$; #$P < 0.05$; ##$P < 0.01$; ns, not significant (ANOVA). **c** Kaplan–Meier survival curves of xenotransplanted mice with shTHBS1-1 (orange line), shTHBS1-2 (red line) or shControl (grey line) transduced P3 cells, based on the presence of neuropathological features ($n = 8$ mice per group); $P$-values were calculated with log-rank test, **$P$-value < 0.01, ***$P$-value < 0.001. **d** P3 cells were transduced with pLA57-control or pLA57-THBS1 lentiviral constructs. Analysis of THBS1 and Tubulin expression in protein extracts from pLA57-control or pLA57-THBS1 P3 cells by western blot (Upper panel) and by immunostaining (lower left panel), THBS1 (green) and F-actin (red) and nuclei (blue). Scale bar: 10 µm. pLA57-control or pLA57-THBS1 P3 cell spheroid invasion was measured in collagen I gels after 24 h. The graph represents the results as means ± s.d of three independent experiments each done in 6–8 replicates for each condition, **$P < 0.01$ (Student t-test)

---

analysis of the RNA-sequencing data confirmed distinct transcriptional profiles for central and invasive tumour areas (Supplementary Fig. 10A). Differential expression analysis revealed gene signatures comprising 520 genes that were higher expressed in invasive areas and 1035 genes that were higher in central areas for P3 tumours (Supplementary Fig. 10B and Supplementary Data 2). Functional analysis of the gene signatures revealed enrichment of Gene Ontology terms being highly relevant to glioma biology (Supplementary Data 3). As expected, central areas showed higher expression of multiple genes related to blood vessel development, cell adhesion, inflammation, cell migration and neurogenesis, whereas invasive areas expressed higher levels of genes controlling chemotaxis, cell migration and cell communication. To further assess potential cellular networks involved in these processes, we used Ingenuity Pathway Analysis to search within the invasive and angiogenic gene signatures for potential secreted molecules involved in cell communication (Supplementary Data 2). This analysis identified TNF and TGFB1 as the most significantly overlapping regulators with 47 and 36 targets, respectively. As previously mentioned, TGFB1 is known to control many processes of GBM development, and interestingly 20 out of its 36 targets were expressed higher in invasive areas (Supplementary Fig. 10C). While focusing on the invasive area, we further compared overexpressed genes that may contribute to communication between the stromal compartment and the tumour. Tumour Thrombospondin-1 (THBS1), Annexin II (ANXA2) and PDGFB were found to have the highest connectivity with genes of the stromal compartment (Fig. 7 and Supplementary Data 3). THBS1 has high connectivity in the invasive compartment of P3 tumours with collagens, TGFB1 and integrins (Fig. 7). The significance of the role of THBS1 is supported by an independent laser-capture microdissection and qPCR analysis, which showed increase of THBS1 mRNA in the invasive area when compared to the tumour core (Fig. 4b). These data are in agreement with the functional analysis described above and, thus, support a role of THBS1 in tumour cell invasion in GBM.

## Discussion

GBMs represent one of the most challenging tumours to treat based on their location and their invasive behaviour in the brain. These tumours also display an extensive cellular heterogeneity. Therefore, elucidating the mechanisms of tumour progression is important for the development of new and effective therapies.

In this article, we focused our attention on THBS1 to determine its involvement in GBM development. We showed that (1) THBS1 expression is greater in high-grade glioma patients

samples when compared with low-grade gliomas; (2) THBS1 expression is regulated by TGFβ1 via SMAD3-binding sites; (3) THBS1 is both expressed in tumour cells and vessels; (4) tumour-derived THBS1 is involved in GBM invasion and expansion; (5) anti-angiogenic treatment increases THBS1 expression through hypoxia-induced TGFβ1; (6) tumour cell-bound CD47 is involved in THBS1 effects; (7) network analysis demonstrates THBS1 as the gene with the highest connectivity in the invasive compartment.

The thrombospondin family has been shown to be involved in tumour development and progression in several types of cancer[29]. THBS1 has been shown to impair GBM growth and vascularisation[30]. Since THBS1 is a matricellular protein with a plethora of regulatory functions, we set out to further elucidate its role through a systematic analysis using multiple cell models and assays with a focus on tumour cell invasion.

A classical paradigm for TGFβ activation relies on the idea that stretching of the TGFβ-LAP protein by THBS1 is involved in this process[24]. However, in our study, treatment of cells with purified THBS1 did not increase fibronectin expression and did not lead to smad phosphorylation in the reporter assay. We then knocked down endogenous THBS1 in cells that highly over-express this molecule and also produce significant amounts of active TGFβ1. This did not alter active TGFβ1 concentrations in the medium, which indicates that THBS1 does not activate TGFβ1 in this case. We therefore sought to determine whether, on the contrary, TGFβ1 was able to regulate THBS1 expression in GBM. THBS1 expression has been reported to be regulated by FGF, PDGF or E2F1[31], but only a few reports exist to support regulation of THBS1 by the TGFβ pathway[32,33]. In our study, only TGFβ1 but not TGFβ2, EGF, PDGF-BB or IL10 stimulated THBS1 expression. TGFβ1-mediated induction of THBS1 is regulated through SMAD3 phosphorylation and nuclear translocation. To obtain insights into the transcriptional regulation, we mutated the two SMAD3-binding motifs in the *THBS1* promoter. One is located at −1125/−1115, whereas the second corresponds to a −110/−100 site. We demonstrated that these two domains have opposite effects on *THBS1* transcription. The first one activates THBS1 transcription while the second one drives inhibitory signals. The net result of THBS1 activation by TGFβ1 may be due to the ratio of Smad binding to site 1 and 2 where binding to site 2 will be favoured. The existence of different target sites has been published for E2F1 with one activator and one repressor target sequences[34]. The control of THBS1 transcription may involve more SMAD3 interactors for defining this positive and negative effects. Further studies on *THBS1* gene regulation will be performed for elucidating this phenomenon.

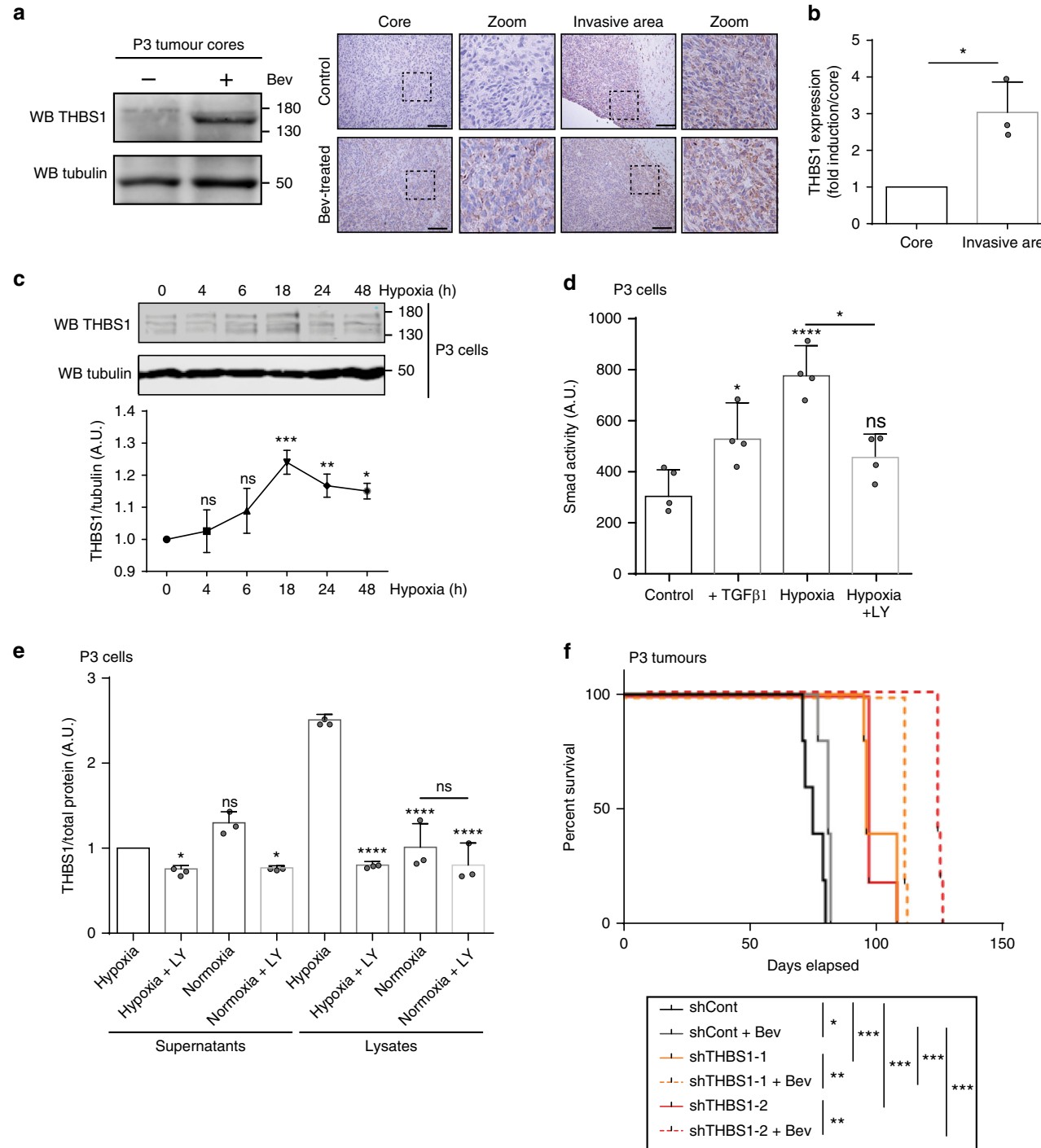

Our data are in apparent contradiction with the results of Seliger and collaborators[35]. These authors reported that THBS1 expression is increased by lactate and that this in turn contributes to TGFβ2 activation. Our data are completely opposed to these published results and one can only speculate about the reason of these differences. TGFβ activation was only studied by the authors in the HTZ-349 cell line and not any other standard or patient-derived glioma cell lines. The authors also studied TGFβ2 activation and not TGFβ1 activation. On the contrary, our study provides robust data that are highly consistent in standard and patient-derived cell lines.

As noted in this article, we have used the U87 cell line for a number of functional experiments. Recently, a doubt has been

cast on the origin of this cell line[36]. Furthermore, U87 cells classically grow as a very compact and round angiogenic tumour in vivo. In our study, we have used both U87 cells and cells from patient-derived tumours. The results derived from TGFβ1 and THBS1 functional analysis were consistent for both U87 and patient-derived cell lines. In addition, we analysed another U87-derived cell line, U87 IRE1dn, which have very high THBS1 levels and a pronounced invasive phenotype in vitro and in vivo[23].

Experiments carried out on the U87 cell line showed impairment of in vitro invasion after THBS1 silencing. In vivo, mouse survival was increased in animals xenografted with U87 THBS1-knockdown tumours when treated with anti-angiogenic therapy. This is also consistent with the results obtained with the P3

**Fig. 4** THBS1 is induced by hypoxia through TGFβ1 activation. **a** Protein extracts from P3 tumour cores were analysed by western blot probed with anti-THBS1 and anti-Tubulin antibodies (left panel). Representative images of P3 control or bevacizumab-treated tumours stained for THBS1 (brown) and counterstained with haematoxylin (blue) are shown (right panels). Images represent the core and the invasive areas of both tumours. Scale bars: 100 μm. **b** qPCR of *THBS1* transcript from the core or the invasive area of P3 tumours, after laser microdissection. Three independent tumours were analysed, results are represented as means ± s.d and compared to the tumour core. *$P < 0.05$ (Student *t*-test). **c** Immunoblots of protein extracts from P3 cells exposed to hypoxia (1% $O_2$) for 4, 6, 18, 24 or 48 h, and probed with anti-THBS1 or anti-Tubulin antibodies. The graph below represents ratios of THBS1 and Tubulin signal intensities. Results are represented as means ± s.d. of three independent experiments. *$P < 0.05$; **$P < 0.01$; ***$P < 0.001$; ns, non-significant (Student *t*-test, comparison to time 0). **d** P3 cells were incubated under normoxia (21% $O_2$) or hypoxia (1% $O_2$) for 48 h and stimulated or not with 5 ng/ml of recombinant TGFβ1 or 10 μM of LY2157299. Cells were transfected with SRE construct and luciferase activity was assessed. Results are represented as means ± s.d. of four independent experiments. **e** ELISA experiments performed on P3 cell lysates and supernatants. Cells were treated or not with 10 μM of TGFβR inhibitor LY2157299, in normoxic or hypoxic conditions for 48 h. The graph represents three independent experiments as fold induction in comparison to non-treated cells (mean ± s.d.). *$P < 0.05$; **$P < 0.01$; ***$P < 0.001$; ****$P < 0.0001$; ns, non-significant (ANOVA). **f** Kaplan–Meier survival curves of mice with intracranially implanted control (grey lines), THBS1-1 (orange lines) or THBS1-2 (red lines) shRNA-transduced P3 cells. Animals were treated or not with Bevacizumab (Bev) ($n = 7$ mice per group); *P*-values were calculated with log-rank test; *$P$-value < 0.05; **$P$-value < 0.01; ***$P$-value < 0.001

tumour model where THBS1-knock down in combination with bevacizumab treatment led to a marked inhibition of both in vitro and in vivo invasion, with a significant impact on mouse survival. A decrease in tumour mass was also observed under these conditions. Taken together, these data indicate that THBS1 has a role in both tumour expansion and invasion. THBS1 inhibition reinforces the effect of anti-angiogenic treatments.

Bevacizumab treatment is known to decrease neo-angiogenesis and restores, to some extent, the blood–brain barrier[37]. This therapy may also induce evasive resistance through upregulation of pro-invasive molecules via HIF1α expression[3]. Hypoxia has previously been shown to induce THBS1 expression[9,38], but the mechanistic link was not established in these studies as no HIF1α binding sequence is present in the *THBS1* promoter. Indeed, our data showed that hypoxia leads to TGFβ1 activation and nuclear accumulation of Smad transcription factors in GBM, which is consistent with the observation that the TGFβ1 promoter exhibits a hypoxia-responsive element[39]. Furthermore, as described above, we demonstrated mechanistically how Smad3 induces THBS1 expression.

A mimicking peptide based on the CD36-activating sequence of THBS1 was developed (ABT-510), but had disappointing results in clinical trials[40]. Other molecules are, at present, under development[28]. One of those, the TAX2 peptide, is aimed at targeting the interaction between THBS1 and CD47. CD47 is expressed in endothelial cells, macrophages, and also in tumour cells[41]. This finding that was confirmed in our glioma cell lines and tumours models. TAX2 as a stand-alone treatment showed inhibitory activity in vitro under hypoxia in P3 cells and impaired single-cell invasion in vivo. TAX2 alone also reduced vascular density to some extent. TAX2 combined with bevacizumab led to an inhibition of contra-lateral tumour invasion in the same tumour model. This indicates that the CD47–THBS1 interaction has a complex role in GBM development by acting on angiogenesis, tumour invasion and expansion. These data are reinforced by CD47 knock down in glioma cells, which demonstrated inhibition of migration, invasion and an increase in survival. Our results are in agreement with recently published data where the authors identified cell-surface signatures for GBM using a systems biology approach[42]. The authors showed that CD47, besides other regulatory molecules, exhibited positive enrichment after TGFβ treatment and that knock down of CD47 inhibited U87 migration. Furthermore, a computational model of THBS1 regulation has been recently established that links hypoxia-TGFβ-THBS1. This also supports our experimental work[43].

To reinforce our data, we choose a non-biased approach by performing RNA-sequencing of central and invasive areas in the intracranial P3 model. Firstly, we identified several important players for GBM development including TGFβ1. Most strikingly, *THBS1* was among the most connected genes in our analysis in the invasive area, which supports the functional analysis described above.

Taken together, our results indicate that THBS1 has an important role in the development of GBM and that a reversed signalling pathway from TGFβ1 to THBS1 is involved. THBS1 may not only act in the angiogenic core of the tumour but also in areas of tumour cell invasion. Furthermore, tumour cell-bound CD47 is important for THBS1 activity in tumour cell invasion (Fig. 8). Finally, THBS1 inhibition may be therapeutically important for reinforcing the efficacy of current anti-GBM treatments by not only acting at the vascular compartment, but also on invasive processes in the tumour periphery.

## Methods

**Ethical issues**. Male RAGγ2C$^{-/-}$ mice were housed and treated in the animal facility of Bordeaux University ("Animalerie Mutualisée Bordeaux"). All animal procedures have been done according to the institutional guidelines and approved by the local ethics committee (agreement number: 4611).

**Patient material**. Patients with preoperative MR imaging studies suggesting a presumptive diagnosis of GBM (defined as intracerebral mass lesion with contrast enhancement area at post gadolinium T1-weighted MR) were included in the study. Volumetric FLAIR and post gadolinium T1-weighted images of these patients were obtained in the preoperative period and loaded into the neuronavigation plan (Brainlab workstation) to be available during surgery for intraoperative tracking. At surgery, various samples were collected at the peripheral region of the tumour (invasive area defined as FLAIR abnormalities around and distant from post gadolinium T1 contrast enhancement ring) or at the tumour core (defined as contrast enhancement area at the tumour ring in post gadolinium T1-weighted images). The site of collection was registered every time on the neuronavigation system and data were stored and collected for subsequent analysis. Part of the samples was immediately frozen and stored in the tumour bank. Another part of the samples was sent for routine histological and molecular analysis (IDH1 mutation, MGMT promoter methylation status, 1p/19q codeletion, ATRX expression). Only patients with a histological diagnosis of GBM and IDH1wt profile were included. Patients gave their consent prior tissue analysis according to the clinical guidelines. Informed written consent was obtained from all subjects (Department of Neurosurgery, Humanitas, Milano according to Humanitas ethical committee regulations, or from the Haukeland Hospital, Bergen, Norway). The patient molecular profile is detailed in Supplementary Table 1.

**Intracranial tumour xenografts**. U87, P3 spheroids were stereotactically implanted into the brains of randomly chosen Ragγ2C$^{-/-}$ mice (8–12 weeks old). Briefly, GBM spheroids (5 spheroids of $10^4$ cells per mouse) were implanted into the right cerebral cortex using a Hamilton syringe fitted with a needle (Hamilton, Bonaduz, Switzerland) and following the procedure already described[44]. Mice were treated with 10 mg/kg (mouse weight) of bevacizumab, or 10 mg/kg (mouse weight) of TAX2, LSKL or control peptides. Mouse survival was based on the presence of neuropathological features (body position, fur quality, eye position) and weight. A minimum of 7 mice per group was chosen to yield enough statistical power ($P = 0.05$), except for one experiment in which 5 mice per group were used (Fig. 5d).

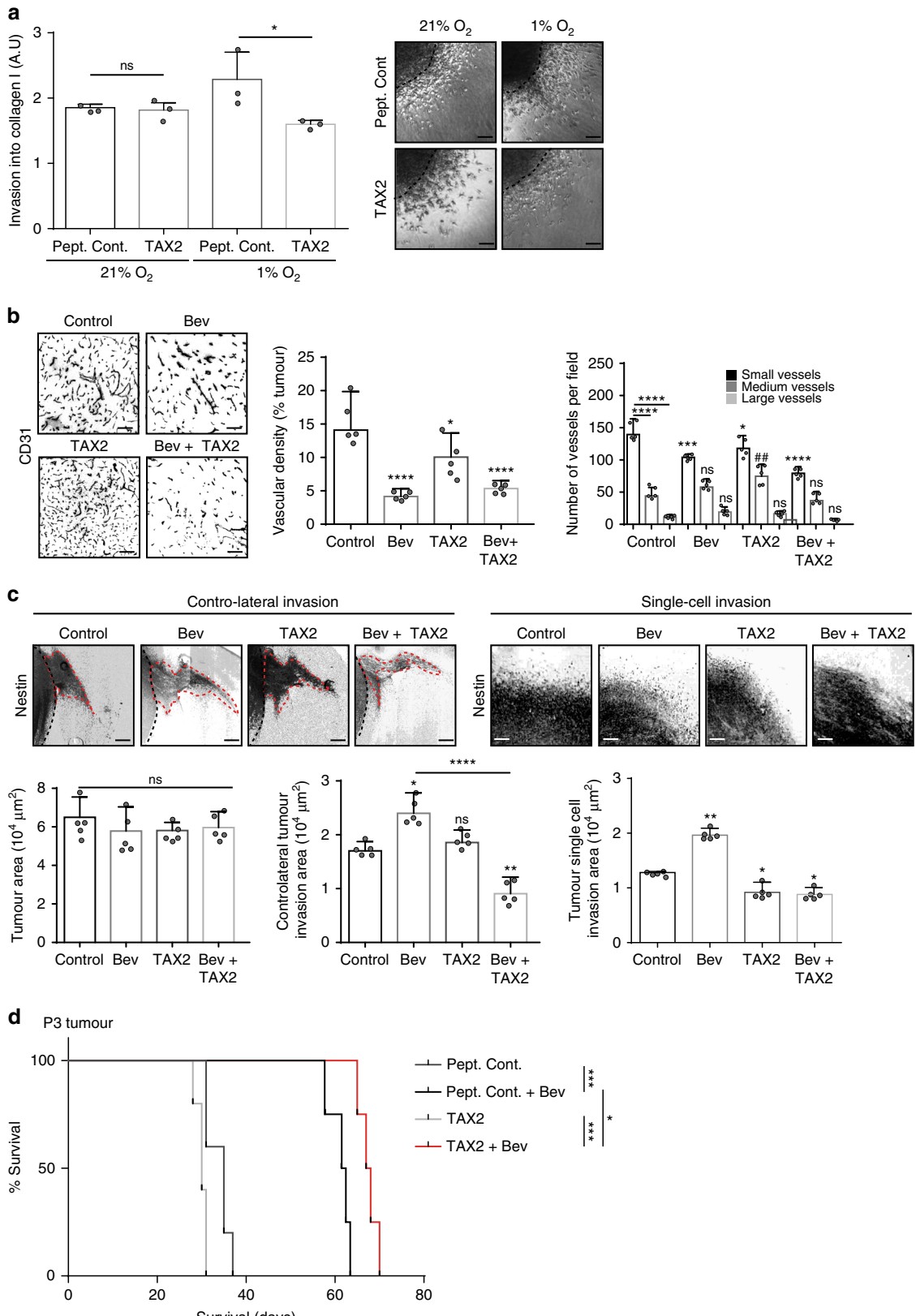

**Reagents and antibodies**. The detailed information of all antibodies (primary and secondary) used in this study is listed in Supplementary Tables 4 and 5. The sequences of siRNA Smad3 are listed in Supplementary Table 6. The primer sequences used for RTqPCR are listed in the same table. TGFβ1 recombinant protein was purchased from Peprotech (100-21), purified THBS1 (R&D System,

3074-TH-050), recombinant TGFβ2 (Peprotech, 100-35B), EGF (Peprotech, AF100-15), PDGF-BB (Sigma, SRP3138) and IL-1b (RD Systems, 201-LB/C). TAX2 peptide (CEVSQLLKGDAC) and scramble control peptide (LSVDES-KAQGIL) were synthesised and purified by Genecust (Dudelange, Luxembourg) and controlled for composition and purity through electrospray ionisation-mass

**Fig. 5** THBS1/CD47 interaction in P3 tumour invasion and growth. **a** P3 cells were included into collagen I gels and then incubated in normoxia (21 % $O_2$) or hypoxia (1 % $O_2$). P3 spheroid invasion was measured in collagen I gels after 24 h. Scale: 50 μm. The graph represents the results as means ± s.d. of three independent experiments, each done in 6–8 replicates for each condition. *$P < 0.05$; ns, non-significant (ANOVA). **b** P3 tumours were treated with 10 mg/kg (BW) of bevacizumab (Bev), control peptide (control), TAX2 alone (TAX2) or in combination (Bev + TAX2). Tumours sections were stained with anti-CD31 antibody. Scale: 20 μm. The graphs below represent the vascular density (left panel) and the quantification of vessel types according to size (right panel: small vessels < 10 μm; medium size vessels between 10 and 20 μm; large vessels > 20 μm) (average of 5 tumours with 8 sections/tumour analysed). Results are represented as means ± s.d. *$P < 0.05$; ***$P < 0.001$; ****$P < 0.0001$ for small vessel comparison; ##$P < 0.01$ for medium vessel comparison; ns, non-significant (ANOVA). **c** P3 tumours were stained for Nestin (grey) to evaluate contra-lateral or single-cell invasions (black dashed lines around tumour edges and red dashed lines around invasive areas). Scale: 100 μm. The graphs below represent core (left panel), contra-lateral invasion (medium panel) and single-cell invasion (right panel) areas of tumours from treated and untreated mice (average of 5 tumours with 8 sections/tumour analysed). Results are represented as means ± s.d. *$P < 0.05$; **$P < 0.01$; ****$P < 0.0001$; ns, non-significant (ANOVA). **d** Kaplan–Meier survival curves of P3 xenotransplanted mice treated with control peptide, TAX2 and bevacizumab alone or in combination ($n = 5$ mice per group); $P$-values were calculated with log-rank test; *$P$-value < 0.05; ***$P$-value < 0.001

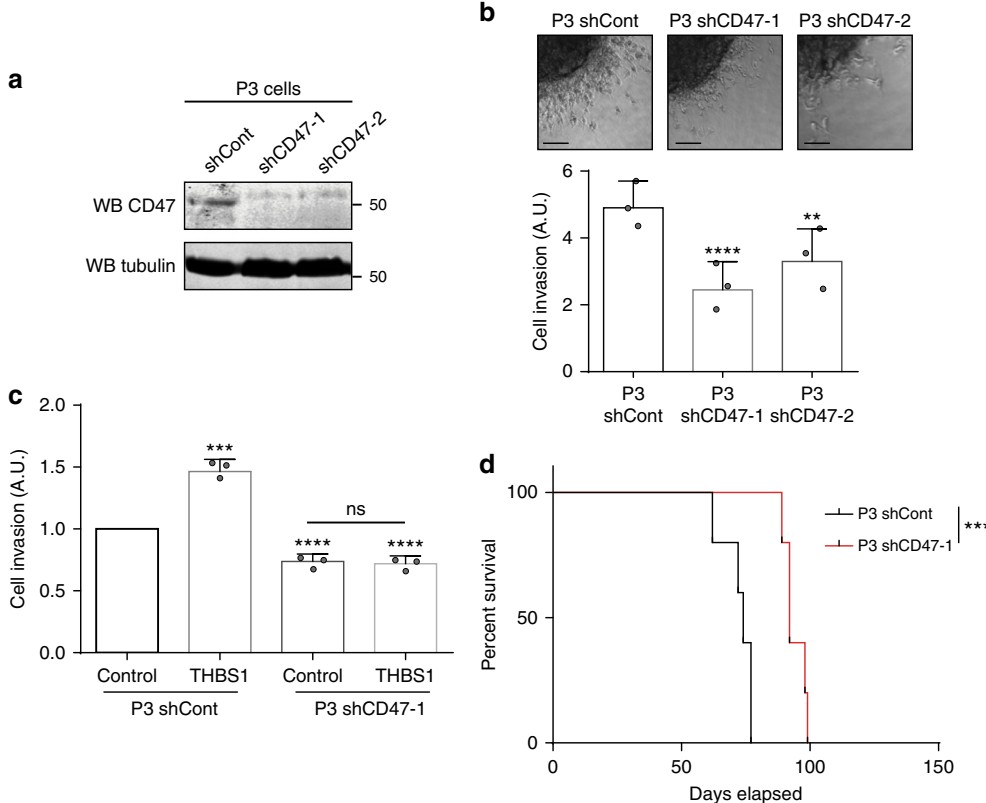

**Fig. 6** Tumour-associated CD47 controls glioma cell invasion and motility. **a** Immunoblots of protein extracts from control or CD47-1/-2 shRNA-transduced P3 cells probed with anti-CD47 or anti-Tubulin antibodies. **b** P3 cells were transduced with control or CD47 (−1 and −2) shRNAs. P3 spheroid invasion was measured in collagen I gels after 24 h. Scale: 50 μm. The graph below represents the results as means ± s.d. of three independent experiments each done in 6–8 replicates for each condition. **$P < 0.01$; ****$P < 0.0001$ (ANOVA). **c** Effect of THBS1 on P3 cells transduced with control or CD47-1 shRNAs. Full-length THBS1 was used at a concentration of 10 μg/ml and invasion in collagen I gels was measured after 24 h. The results are expressed as means ± s.d. of three independent experiments each done in 6–8 replicates for each condition. **$P < 0.01$; ****$P < 0.0001$; ns, non-significant (ANOVA). **d** Kaplan–Meier survival curves of mice bearing control or CD47-1 shRNA-transduced P3 tumours ($n = 5$ mice per group); $P$-values were calculated with log-rank test; ***$P$-value < 0.001

spectrometry (ESI-MS) and HPLC. Peptides were solubilized in the appropriate culture medium for in vitro assays (DMEM or Neurobasal medium), or in normal saline solution (0.9% (w/v) NaCl) for in vivo experiments. Cells were treated with 10 μg/ml of Tunicamycin (Sigma) or 10 μM of LY2157299 (TGFβ receptor I inhibitor or Galunisertib). Animals from the treatment group received one weekly i.p. injection of bevacizumab (10 mg/kg) every 2 days all along the experiment.

**Laser capture microdissection and RNA-sequencing.** Coronal brain sections (30 μm thickness) were made using a CM3050-S microtome (Leica, Wetzlar, Germany) and mounted on 1 mm polyethylene-naphthalate membrane glass slides (P.A.L.M. Microlaser Technologies AG, Bernried, Germany) that have been pretreated to

inactivate RNase. Subsequently, sections were, dehydrated in a series of pre-cooled ethanol baths (40 s in 95% and twice 40 s in 100%) and air-dried. Laser microdissection of samples was performed using a PALM MicroBeam microdissection system version 4.6 equipped with the P.A.L.M. RoboSoftware (P.A.L.M. Microlaser Technologies AG, Bernried, Germany). Laser power and duration were adjusted to optimise capture efficiency. Microdissection was performed at ×5 magnification. Four tumours were analysed for each condition, and five caps were collected for each tumour type. Samples were collected in adhesives caps and resuspended in a guanidine isothiocyanate-containing buffer (RLT buffer from RNeasy minikit, Qiagen, Chatsworth, USA) with 10 μl/ml β-mercaptoethanol, and stored at −80 °C until extraction was done. Total RNA was extracted from microdissected tissues using the RNeasy® mini Kit (Qiagen, Hilden, Germany) according to the

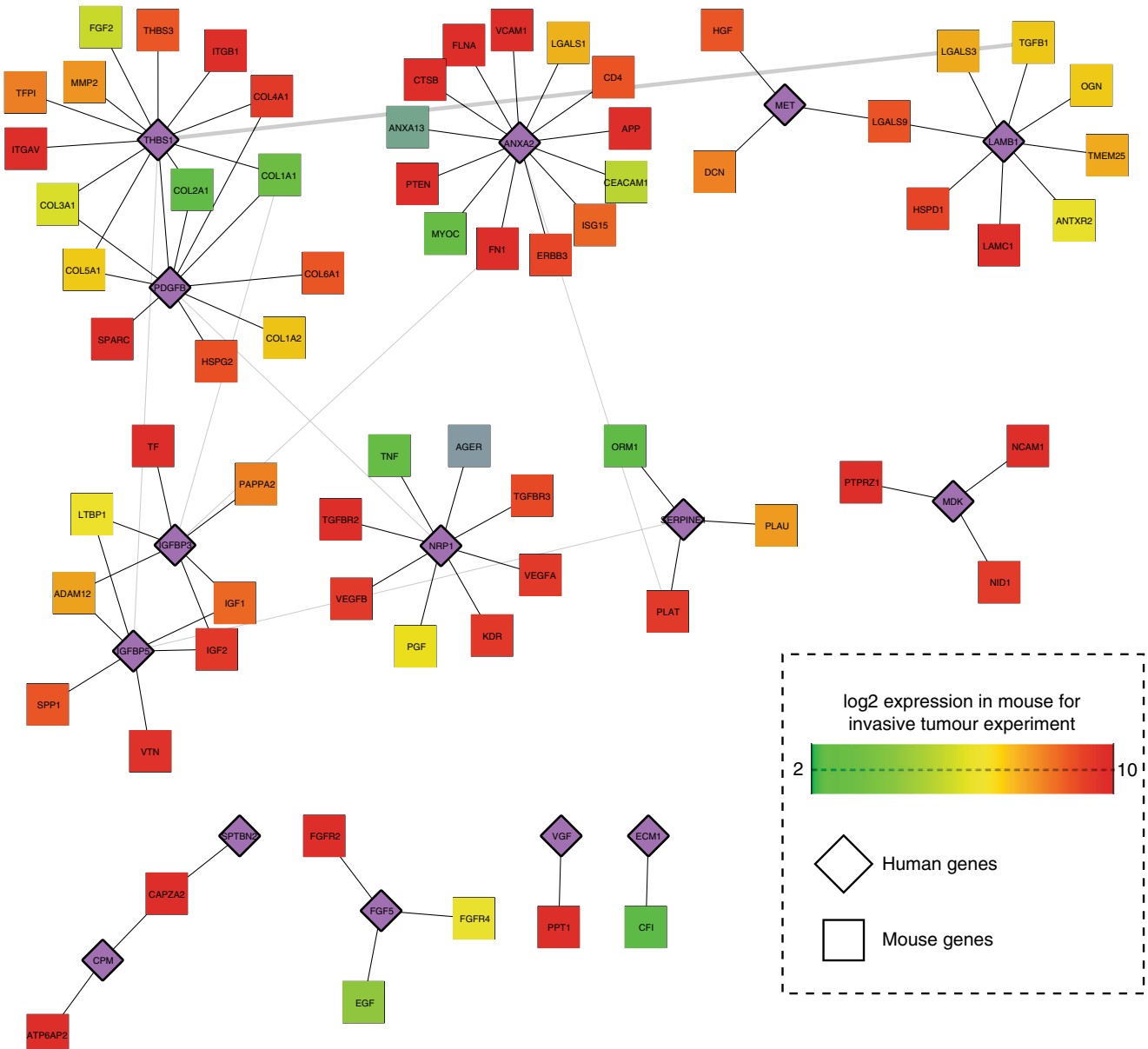

**Fig. 7** Transcriptional analysis highlights THBS1 as central regulator of GBM invasion. A network of proteins encoded by genes that are overexpressed in the invasive area of P3 tumours when compared to the core area, or are expressed by the host. Relationships in the network represent known protein–protein interactions. Human genes are represented in purple and mouse genes are colour coded according to their expression (green (2) – red (10); normalised counts in log2 scale)

manufacturer's protocol. The integrity of the RNA was checked by capillary electrophoresis using the RNA 6000 Pico Labchip kit and the Bioanalyser 2100 (Agilent Technologies, Massy, France), and quantity was estimated using a Nanodrop 1000 (Thermo Scientific, Waltham, USA). The RNA integrity number (RIN) above 7/8.

**RNA-sequencing for P3 tumours**. Overall, 500 ng was used as input material for a Low Input Ribozero treatment using the Epicentre Ribo-Zero Gold Kit–Low Input (HMR), and subsequently purified using Ampure XP beads. The samples were then analysed on the Agilent 2100 Bioanalyzer and all of the depleted RNA was used as input material for the ScriptSeq v2 RNA-Seq Library Preparation protocol. Following 14 cycles of amplification, the libraries were purified using Ampure XP beads. Each library was quantified using Qubit, and the size distribution assessed using the Bioanalyzer.

These final libraries were pooled in equimolar amounts using the Qubit and Bioanalyzer data. The quantity and quality of each pool was assessed by the

Bioanalyzer and subsequently by qPCR using the Illumina Library Quantification Kit from Kapa on a Roche Light Cycler LC480II according to manufacturer's instructions.

The template DNA was denatured according to the protocol described in the Illumina User guide and loaded at 12.5 pM or 13 pM concentration. The sequencing was carried out on two lanes of an Illumina HiSeq2500 at $2 \times 125$ bp paired-end sequencing with v4 chemistry.

**Bioinformatics analysis**. Reads quality filtering, species assignment and genomic alignment: Read quality check and filtering has been performed using NGS QC Toolkit v.2.3.3[45] with default parameters. Less than 1% of the reads has been discarded at this step. Unpaired reads have been kept and represent 0.005% of the data. In order to assign the reads to Human and Mouse, we have used Xenome v.1.0.1[46]. It is a two-step process. During the first step, we created the genome index files built from EMBL release 84 (Human GRCh38 and Mouse mm10). We used the index files to classify the sequence as host or graft separately for paired

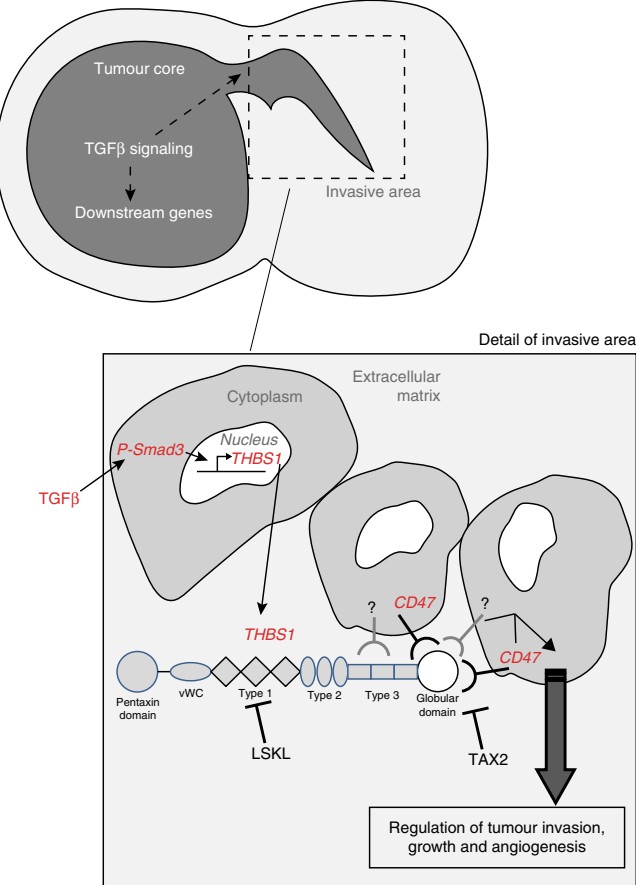

**Fig. 8** Proposed model for GBM invasion TGFβ1 is expressed in both the core and the invasive areas in GBM. THBS1 is transcriptionally regulated via SMAD3, which binds to regulatory elements in the *THBS1* gene. THBS1 will then be released and act on tumour cell invasion and expansion. The interaction with CD47 is critical in this process

and single reads. The genomic alignment was performed with Bowtie2[47] with default parameters on the same reference genomes as used for Xenome. Reads with quality mapping score lower than 10 were discarded.

Differential expression between the tumour core and invasive areas: We first assigned reads to hg38 (human) and mm10 (mouse) gene models retrieved from[48]. Using the R packages GenomicFeatures and GenomicAlignments[49], 25,446 and 24,423 genes were analysed in human and mouse respectively. In order to compute differential expression, we performed two-class comparisons between tumour core and invasive samples using the R package DESeq2[50]. Differentially expressed genes were defined by using an upper threshold on the *P*-value of 0.01 after correction for multiple testing using the method of Benjamini and Hochberg[51].

Functional enrichment: The R packages ClusterProfiler[52] and DOSE[53] were used to identify functional enrichment in groups of genes in human and mouse. These packages give access to the Gene Ontology (GO) categories (molecular function, biological process, cellular component), KEGG pathways, the Disease Ontology (DO) and the Network of Cancer Genes (NCG).

Network building: Protein–protein interaction networks representing putative signalling events occurring in the invasive tumour microenvironment/ECM were constructed as followed (i) Mouse expression data were annotated using human orthologs of mouse genes retrieved from EnsEMBL release 85 and used for subsequent analysis (www.ensembl.org/). (ii) Host and tumour data were subset for genes that were associated to Gene Ontology terms Extracellular Space (GO:0005615), Extracellular Region (GO:0005576) and External side of plasma membrane (GO:0009897). (iii) Genes expressed by the host were determined as present in the extracellular compartment if their average expression was higher than 2. (iv) Genes expressed in tumours and overexpressed in invasive regions were selected from the differential expression analysis. (iv) Non-redundant interactions between proteins represented by the gene sets identified in (iii) and (iv) were retrieve from BIOGRID v. 3.4.140[54]. (v) Visualisation of the resulting network was performed using Cytoscape[55] and the community clustering algorithm GLay within the Clustermaker plugin (www.cgl.ucsf.edu/cytoscape/cluster/clusterMaker.shtml).

THBS1/TGFB1 expression between GBM grades: Analysis of THBS1 and TGFβ1 expressions in gliomas of various grades was performed using data from Sun et al.[56]. The fold changes are represented in log2 scale. This gives a log2 fold change of $8.59 - 6.32 = 2.27$. To convert this to a linear fold change, we calculate 2 to the power of 2.27 ($2^{2.27} = 4.82$).

**Cell culture and transfection**. All cells used for phenotypic and functional studies have been further characterised more in detail by cGH array (P3) and by cell authentication (U87) using Promega Powerplex21 Kit (Eurofins, GE). The U87 (ATCC) and P3 glioma cell lines were regularly tested for contamination and were all mycoplasma free. Cells were cultured in Dulbecco's Modified Eagle's medium (DMEM, Thermo Fisher Scientific) supplemented with 10% Fetal Bovine Serum (FBS), 5% antibiotics (Penicillin and Streptavidin) and 5% L-glutamine. U87 cells were cultured at 37 °C, 5% $CO_2$ and split at 70–90% of confluence with 0.25% Trypsin. P3 spheroids were extracted for P3 tumours and cultured in agar-layered flasks for few weeks. The patient-derived cell lines P3, BL9 and BL13 were isolated by enzymatic digestion from P3 tumours or from patient samples (BL9 and BL13) and then cultured in Neurobasal Medium (NBM, Thermo Fisher Scientific) supplemented with B27, 0.2% heparin (100 U) and 20 ng/ml basal FGF. Cell transfection was performed using GeneCellin Reagent (Eurobio) according to the manufacturer instruction, for luciferase plasmids (see Methods below). Cells were also cultured in hypoxic conditions (1% $O_2$).

**Small interfering RNA knockdown experiments**. For transient inhibition of Smad3 expression, cells were transfected with annealed siRNA. Transfections were performed using lipofectamine RNAimax kit (Fisher 10601435) in accordance with the manufacturer protocol, with a final concentration of 20 nM in media without antibiotics. After transfection, cells were washed twice with PBS and fresh complete media was added for 48 h.

**shRNA constructs, lentiviral production and cell infection**. The shRNA target sequences for THBS1 and CD47 are listed in Supplementary Table 6. Gain-of-Function experiments were assessed by overexpressing full-length human THBS1 in a pcDH1 lentiviral vector (Invitrogen). The lentiviral particles were produced by transfecting HEK293T with pMD2.G (VSV-G, viral envelope) and psPAX2 (packaging construct) with 20 µg of bacterial plasmids. Viruses were collected after 48 h of transfection and then filtered (0.22 µm filter). Infectious particles were titrated with scale infections of U87 cells.

**RNA extraction, semi-quantitative and quantitative RT-PCR**. Total RNA was extracted from cells using the TriZol reagent protocol (Invitrogen). After quantification, reverse transcription was done with 2 µg of total RNA using the High Capacity cDNA Reverse Transcription kit (Applied Biosystems). Human THBS1, human CD47 and β-Actin mRNA expression levels were measured by real-time quantitative polymerase chain reaction (qRT-PCR) using Power SYBRGreen PCR Master Mix (Applied Biosystems) on StepOne (Applied Biosystems), according to the manufacturer's protocol. Primer sequences are listed in Supplementary Table 7. The relative abundance of transcripts was calculated by using β-actin transcript quantity as standard. The quantitative RT-PCR experiments were carried out in triplicate on RNA isolated from three independent cell cultures.

**Promoter and luciferase reporter assay**. Wild-type THBS1 promoter was amplified by PCR using platinum pfu (Invitrogen), sequenced and cloned in pGL3-basic vector at *Sac*I and *Nhe*I sites. Del1 and Del2 mutants were amplified by PCR, sequenced and cloned at the same sites. Luciferase promoter activity was measured by quantifying Renilla and Firefly Luciferase activities with the microplate reader Infinite F200 Pro (TECAN) using the Dual-Glo Luciferase Assay System (Promega), according to the manufacturer's instructions. U87 cells were transfected with the pRL-TK vector and either with plasmids carrying the full-length promoter of THBS1, or Del1 or Del2 mutants for promoter studies.

For SMAD reporter assay, cells were transfected with SBE4-luc vector containing SMAD responsive elements. For promoter studies, a control pEGFP plasmid was used to assess transfection efficiency. The pRL-CMV vector (Promega) allowed control and normalisation of the transfection. All transfections and measurements were performed in five different wells and repeated at least three times.

**Western blot**. Two procedures were used for protein lysis, depending of the proteins of interest. (1) Cells were washed twice with phosphate-buffered saline (PBS) and dissolved in lysis buffer (10 mM Tris-HCl, pH 7.4, 150 mM NaCl, 0.5% NP-40, 1%TritonX-100, 1 mM EDTA) supplemented with protease and phosphatase inhibitors cocktails (Roche). Protein concentration was quantified by Bradford assay (Euromedex). Cell lysates were resuspended in Laemmli Buffer (62.5 mM Tris pH 6.8, 10% glycerol, 2.5% SDS, 2.5% β-mercaptoethanol). (2) Cells were washed twice in PBS and directly lysed in Laemmli Buffer. All protein extracts were then boiled for 5 min and separated by SDS–PAGE. Proteins were electroblotted onto a polyvinylidene difluoride (PVDF) membranes (Fisher Scientist, 10344661). Membranes were incubated with blocking buffer (EuroBio 927-40003)

for 1 h, probed overnight at 4 °C with the primary antibodies of interest, followed by detection using secondary antibodies coupled to Fluoroprobes (Supplementary Table 5) and Odyssey infra-red imaging system (Li-Cor Biosciences, Nebraska, US). Densitometry analysis was performed using Fiji software. Uncropped immunoblots are depicted in Supplementary Figures 11 and 12.

**Human THBS1 ELISA.** The Quantikine ELISA for Human THBS1 (R&D Systems) was used according to the manufacturer's protocol. Proteins from lysates were extracted with 10 mM Tris-HCl, pH 7.4, 150 mM NaCl, 0.5% NP-40, 1% TritonX-100, 1 mM EDTA supplemented with protease and phosphatase inhibitors cocktails (Roche). Protein contents from lysates and supernatants were determined by using the BCA protein assay kit (Pierce).

**Indirect immunofluorescence staining.** Cells were seeded on glass coverslips and fixed 10 min with 4% paraformaldehyde at room temperature, then washed in PBS. Cells were permeabilised 10 min with 0.1% TritonX-100, washed in PBS and incubated for 1 h with blocking buffer (PBS containing 2% FBS and 1% BSA) at room temperature. Cells were incubated with primary antibodies diluted in blocking buffer overnight at 4 °C, washed with PBS, and incubated with secondary antibodies diluted in blocking buffer (Supplementary Table 5). DAPI was used to label nuclei (Fisher Scientist, 10374168). Coverslips were mounted using Prolong Gold antifade reagent (Fisher Scientist, 11559306).

**Histological and immunohistological analyses.** Tissues embedded in OCT were cut in frozen sections of 10 µm using the microtome LEICA CM1900. For histological analyses, frozen sections were stained with haematoxylin and eosin. For immunofluorescence on histological sections, frozen sections were processed as described in the indirect immunofluorescence staining section.

Patient paraffin-embedded sections were deparaffinised in xylene and hydrated serially in 100, 95, and 80% ethanol. Endogenous peroxidase was quenched in 3% $H_2O_2$ in PBS for 1 h. Slides were then incubated with antibodies (Supplementary Table 4) overnight at 4 °C. Sections were washed three times in PBS, and antibody binding was revealed using the Ultra-Vision Detection System anti-Polyvalent HRP/DAB kit according to the manufacturer instructions (Lab Vision). Finally, the slides were counterstained with haematoxylin and washed in distilled water. After dehydration and mounting, THBS1 expression localisation was analysed by using IHC profiler from Fiji® Software. For THBS1 immunostaining, tissue microarrays were used to compare patient grade samples (numbers B17016c and B17015a, BioMax, US).

**Invasion assays in collagen I gels.** P3 spheroids were prepared 3 days respectively before inclusion by seeding of $10^4$ cells in neurobasal medium with 0.4% methylcellulose (Sigma) in a U-bottom 96 wells plate (Falcon). A solution of 1 mg/ml of collagen I (Fisher Scientific) was prepared in PBS with 7.2 mM NaOH. After 30 min of incubation on ice, spheroids were individually picked, washed in PBS and included in the collagen solution. After 45 min at 37 °C in a cell incubator, neurobasal medium with the different treatments is added. P3 spheroid invasion areas are measured after 24 h with FIJI software. The total area was delineated by two independent investigators as well as the central spheroid core. An invasive index was calculated by the ratio of the total spheroid surface and the spheroid core.

**Invasion assays on U87 cells.** U87 cells were seeded on 0.2 mg/ml Matrigel coatings into a ImageLock 96 well plate. The day after, wounds were created using Incucyte™ wound maker and a 2 mg/ml Matrigel layer was deposited on top of the cells. The plate was then processed for invasion using Incucyte™ device.

**Quantifications of vascular density, invasive index.** Vascular density: This was estimated as the ratio of the vessel surface (CD31 staining) to the total tumour surface (Nestin staining). Each experiment was analysed with an average of 5 tumours with 8 sections/tumour using Fiji software[57]. Evaluation of vessel length: 3–4 images stained with CD31 were analysed for evaluating blood vessel quality (small vessels < 10 µm length, medium vessels between 10 and 20 µm length, and large vessels > 20 µm length) using counter from Fiji[57].

Invasive index: This was estimated by measuring the surface of the invasive area ($mm^2$). Each experiment was analysed with an average of 5 tumours with 8 sections/tumour using Fiji[57].

**Live-cell microscopy.** To assess cell motility, U87 cells were seeded at a density of $2 \times 10^4$ cells per well on Matrigel coated glass bottom 24-well plates (Greiner), as published before[58]. Hoescht 33342 (10 ng/ml) was added to label the nuclei. Live-cell imaging was performed with a Eclipse Ti Nikon video microscope coupled with NIS analysis software, based on a ×10 NA 0.30 objective lens (Nikon), a Hamamatsu Digital CCD C10600-10B camera, and an environmental chamber to maintain cells at 37 °C in a 5% $CO_2$ humidified atmosphere (Life Imaging Services). Videos were acquired every 15 min for a period of 24 h in brightfield and blue filters (Hoescht 33342). Mean cell displacements were calculated with the ImageJ plugin TrackMate. The software was used to automatically track nuclei and characterise their trajectories among stacks of images.

**Statistical analysis.** Statistical analysis was performed using the Graphpad software. Multiple comparisons were performed with one-way analysis of variance, followed by Tukey post hoc tests and with one-way ANOVA Bonferroni multiple comparison test. Statistical comparison between two groups was performed by using the Mann–Whitney $U$ test.

Survival analysis were analysed by using log-rank test.

**Reporting summary.** Further information on experimental design is available in the Nature Research Reporting Summary linked to this article.

## Data availability
All data are available within the Article and Supplementary Files, or available from the corresponding authors on reasonable request. Transcriptomics data are available from the SRA: PRJNA320047 or the European Nucleotide Archive: PRJEB23786. We have also uploaded supplementary material to the Figshare repository (for e.g.: https://figshare.com/articles/Differentially_expressed_genes_between_core_and_invasion_GBM_areas_RNAseq/7472036).

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

## Acknowledgements

This work was supported by grants from INSERM (recurrent funding), and from the "Ligue contre le Cancer" to A.B. and by Helse Vest, Haukeland Hospital, The Norwegian Research Council and Stiftelsen Kristian Gerhard Jebsen Research Foundation for R.B. T.D. was a recipient of Norwegian Cancer Society. LCM facility is supported by LabEX BRAIN ANR-10-LABX-43 and FRM DGE20061007758. We thank A. Barre for first bioinformatics analysis of RNA-sequencing results. Library generation and sequencing were performed by L. Rainbow, C. Nelson and A. Lucaci, and the bioinformatics work was performed by R. Gregory at The Centre for Genomic Research, University of Liverpool and P. Koldkjaer for performing RNA-sequencing. We also thank M. Delugin, J. Massiere, M. Poulet and T. Chouleur for preliminary experiment, the Bordeaux Imaging Center for Nanozoomer use, M.P. Algeo, M.A. Derieppe, M. Campistron for animal treatments and care.

## Author contributions

T.D., C.L., L.S. and S.G. performed experiments; K.C., F.F. and E.D. performed bioinformatics RNA-sequencing and analysis; R.P. and L.A. implanted animals; L.A. constructed luciferase THBS1 reporter; M.M. performed LCM experiments; M.M., S.B., J.-J.F., S.D., A.J. provided advice and reagents; H.M., M.R. and L.B. analysed patient material; T.D., A.J., S.D., R.B. and A.B. discussed the results; A.B. supervised research; T.D., R.B. and A.B. wrote the manuscript.

## Additional information

**Competing interests:** The authors declare no competing interests.

