## [Peer Review File · Nature Communications]

Reviewers' Comments:

Reviewer #1:

Remarks to the Author:

In this manuscript, Daubon et al. reported that THBS-1 highly expressed in the invasive edge and the tumor vessels promotes GBM tumor invasion. They identified the differential expression of THBS-1 by RNA-seq comparison between cells from invasive areas and angiogenic areas. They showed that TGF β is in the upstream of THBS-1 and induces THBS-1. Binding of the phosphor-SMAD2/3 to THBS-1 promoter may be responsible for the upregulation of THBS-1 by TGF β stimulation. They showed that THBS-1 may inhibit the RhoA inhibitor RND3, and thus activate RhoA to promote cell migration and invasion. In the orthotopic xenograft model, they showed that knock-down of THBS-1 inhibited tumor invasion, extended mouse survival, and suppressed tumor vascularization. Finally they used Bev, THBS-1/LAP inhibitor LSKL, and THBS-1/CD47 inhibitor TAX2 to test the effect of combined treatments on tumor invasion and vascularization. Although the topic of GBM invasion is interesting, the main conclusions of this manuscript are not supported by the current data. The significance of the manuscript is severely compromised by several critical concerns.

Main concerns:

1. The manuscript identified THBS-1 as an invasion-associated gene by comparison of micro-dissected samples from so-called invasive areas and angiogenic areas. How did the authors define the invasive areas and angiogenic areas? Did they specifically choose or avoid vessel cells that should exist in both invasive and angiogenic areas? Aside from the invasive areas and angiogenic areas, are the remaining regions of tumor mass analyzed? The authors should present a representative IHC image for the criteria they applied to get the cells for RNA-seq.
2. It seemed that TGF β was randomly chosen for analysis of the THBS-1 signaling since they proposed that TGF β may be an upstream factor of THBS-1 without any clue. Although they provided some evidence that TGF β may activate the transcript of THBS-1 in P3 cells, their data also showed that TGF β had no effect on THBS-1 in NCH421K cells. From these data, it is not convincing that TGF β is a dominant regulator of THBS-1 in GBMs. The authors should test the expression of other reported upstream regulators of THBS-1.
3. The source of THBS-1 is not clarified. Is it expressed by tumor cells? Or it is expressed by endothelial cells? If it is expressed by endothelial cells, do normal endothelial cells express THBS-1? Meanwhile, although the manuscript showed some evidence of the THBS-1 expression in tumor cells in invasive edge in P3 xenografts, they also showed universal expression of THBS-1 in NCH412K xenografts. Thus, it is possible that THBS-1 is universally expressed in tumor cells.
4. Whereas GBM invasion is surely a critical point for the malignancy of the tumor, due to the rapid tumor growth in mouse models, it is hard to directly link the tumor invasion to the animal survival. Given that the authors also showed that in the long time culture, knock-down of THBS-1 impaired tumor cell proliferation, it is likely that the tumor invasion is not an import issue for the delayed animal death after knock-down of THBS-1 with or without Bev treatment (Fig 5).
5. THBS-1 is reported as an inhibitor for angiogenesis by inhibiting VEGFR2 signaling. But the authors showed that shTHBS-1 dramatically inhibited tumor vascularization. They should interpret the phenomenon with some data.

Minor concerns:

1. The infrared photograph showed very different background signal in normal brain tissues in S.Fig3. This may be due to the position of the sections. However, this means that the infrared method is not very reliable.
2. How did the authors calculate the fold G4 vs G2 in Fig. 2A? The same table showed that the G2 mean is 6.32 and the G4 mean is 8.59.
3. Please enlarge the Fig 2C for grade II and III sections, so that the cell types with THBS-1 staining could be determined.
4. In Fig. 2B, although the 2 grade IV tumor samples had relatively high THBS-1 expression, the difference between these 2 samples was dramatic.
5. For all immunoblot showing p-SMAD2/3, the total SMAD2 and SMAD3 should be shown.

6. Fig. 3D showed no significant difference in the immunoblot.
7. Fig. 3F showed that deletion of SMAD3 binding site#1 significantly increased the activation of THBS-1 promoter even without TGF β stimulation. Please explain.
8. Fig 4D and 4E showed wound healing, which does not represent cell invasion but only cell migration. So the statement in the text was wrong.
9. In Fig.5, the authors regarded Bev treated xenografts as hypoxic samples. They should provide some direct evidence that the samples did contain more hypoxic regions and the THBS-1 expression was in these regions.
10. It is strange that all the invasions the manuscript showed looked like protrusions. Did they observe any scattering spots as the consequence of invasion?
11. What does vascular coverage in Figs. 6 and 7 mean? Do they mean vessel density? How did they calculate it?
12. No vessels can be seen in Fig. 6F.

Reviewer #2:

Remarks to the Author:

Daubon et al. claim that TGF β 1-induced expression of THBS1 increases invasion in murine glioblastoma models. Furthermore, they demonstrate that silencing of THBS1 has an additive therapeutic effect with Bev in an in vivo GBM model. It is although troublesome that the shRNA used in vivo did not induce efficient knockdown in vitro. While the findings are of potential interest, the data provided do not convincingly support the claims of the authors. Moreover, the correlation between THBS1 and THG β is not novel (Seliger et al., 2013).

Although they provide mechanistic in vitro data on the induction of THBS1 by TGF β 1 in U87 cells, there are major concerns regarding their validation in vivo as well as other aspects of the study. As the data differ significantly between the P3 and NCH421K model and also from other published data in other entities (e.g. Fernando et al., Clinical Cancer Research 2008), their implication and reproducibility is unclear. Additional in vivo models and larger patient cohorts showing correlation of THBS1 level and meaningful clinical parameters with sound statistics are necessary to support the main claim of the manuscript. Furthermore, Gain-of-Function approaches with overexpression of THBS1 in vitro and in vivo are desirable to proof the underlying mechanism.

Major comments:

1. Authors should rewrite the title because it appears overstated. From their results they cannot conclude that thrombospondin-1 contributes to TGF β 1-dependent invasion in glioblastoma. Interaction between TGF β 1 and THBS1 is only shown in U87 cells in vitro. More cell lines and in vivo data from different GBM models are necessary to validate this mechanism.
2. It is unclear how authors come to the conclusion that THBS1 is 4.8-fold upregulated in G4 vs. G2 when the mean is 8.6- and 6.3-fold in G4 vs. G2, respectively. This needs to be explained better.
3. Page 6, line 134: The authors state that the P3 and the NCH421K tumor model show different results regarding the expression of THBS1 due to heterogeneity of glioblastoma. Checking two tumor models is not sufficient to support this statement. The authors need to investigate more models. Currently, the results from one model cannot be validated in the other, thus their relevance is unclear.
4. It is not clear which conclusions are drawn from the IR microscopy and how these analyzes relate to THBS1 and the idea of the paper. This is further complicated by the fact that results are not consistent among both tumor models.
5. Figure 2C: The authors should provide histomorphometric quantification and statistical tests instead of representative pictures to quantify THBS1 expression in tumors. The number of analyzed patients should be stated in the figure legend as it was done for figure 2B.

6. The quality of the figures needs significant improvement throughout the manuscript, this applies especially to the immunohistochemistry pictures. Here, also higher magnifications need to be displayed. The differences described in the manuscript for example in figure 2D and E are not evident from the pictures. Also it appears rather uncommon to label Figures as e.g. 2Db.

7. The authors claim that THBS1 expression is highest in peri-necrotic and hypoxic areas but hypoxic areas cannot be characterized by morphological criteria. Tumor border areas were chosen to measure THBS1 signal intensities. Those tumor areas are usually rather well oxygenated instead of being hypoxic. If the authors wish to prove that THBS1 expression predominantly occurs in hypoxic tumor areas, murine tumor models are warranted including injections of the hypoxia marker pimonidazole.

8. Figure 2 E: The authors claim higher THBS1 levels in invasive areas using IHC. It is not clear how authors discriminate between angiogenic and invasive areas as this is not evident from the pictures provided.

9. Figure 2E: It is of concern that the data regarding THBS1 expression is not consistent among both tumor models.

10. It is difficult to understand, why the authors use the cell line U87 in Figure 3, 4 and 5 to study the mechanisms, pathways and biological effects of THBS1 and TGF β 1 while other experiments are performed with P3 or NCH421K cells. Cell lines and models need to be used more consistently in vitro and in vivo.

11. Figure 5A: How was the quantification of THBS1 intensities done - no description is provided in the methods section. Detailed description of how statistical significance was evaluated should be provided.

12. Figure 5F: How can this strong, synergistic effect of Beva + shTHBS1 be explained in relation to the rather small anti-tumor effect of the single treatments? How do the NCH421K and P3 PDX models respond to the combinatorial treatment?

13. Suppl. Fig. 6B: The staining for THBS1 is hardly visible in any of both pictures. Please replace them.

14. Figure 6A and 6D, why does P-SMAD have one band in P3 cells and two bands in NCH421K cells?

15. Suppl. Fig. 9A: The western blot for THBS1 is of poor quality. The bands cannot be appreciated properly. Please replace them.

16. The authors intend to show that THBS1 exerts its function on glioblastoma cells by decreasing RND3 level which subsequently activates RHOA. So far, the data are correlative and descriptive. Authors should provide mechanistic data by Gain- and Loss-of-function experiments.

17. Page 9, line 217 and Figure 4A: Authors state that THBS1 expression was strongly reduced with both shRNAs, however, there is only a marginal reduction of THBS1 expression using shTHBS1-1 (reduction from control is only 16 %). It is therefore hard to understand, why shRHBS1-1 and shTHBS1-2 have the same effect on RND3 expression and wound healing when compared to the shRNA control. This suggests that the observed effects might not be specific.

18. As shTHBS1-1 does not seem to reduce THBS1 expression, it is important that authors specify, which shRNA has been used for THBS1 knock down in the in vivo experiments (Figure 6). Please refine the figure legends accordingly.

19. U87 cells do not seem to express high levels of THBS1. In addition to shRNA mediated knock

down it would therefore be necessary to also overexpress it in U87 cells and study the biologic effect in vitro and in vivo.

20. Suppl. Fig. 7: It is not described in the text or in the figure legend in which cells the experiment has been performed.

21. Suppl. Fig. 8B and C: The images are blurry. Please replace them.

22. It is not sufficient to perform shRNA-mediated knock down experiments with only one shRNA. The in vivo and in vitro experiments need to be repeated with another shRNA against THBS1 to rule out off-target effects.

23. THBS1 has been well-studied and characterized as a potent anti-angiogenic mediator. It is therefore difficult to understand that its absence does not change (Figure 6C) or even decrease (Figure 6 F) vascular coverage of glioblastoma in vivo. This is also not in line with previous data showing re-expression of endoglin upon THBS1 knock down.

24. Bevacizumab is widely used as a therapy in murine glioblastoma models. It is therefore surprising that it does not reduce the tumor load although it was used at common dose levels (Figure 7B and C). This needs to be explained.

25. The authors showed that loss of THBS1 signaling reduces invasion thereby prolonging survival of tumor-bearing mice (Figure 5). It is of concern that THBS1/CD47 blockade with TAX2 does not reduce invasion in both tumor models. This is of special importance as TAX2 treatment does not seem to have an effect on tumor load. Please explain.

26. If the authors want to show that inhibiting THBS1/CD47-signaling holds potential as a therapeutic approach, please provide survival data for tumor bearing-mice treated with Beva +/- TAX2 (e.g. for P3 and NCH421K from Figure 7).

27. Figure 7: Consistent readouts should be provided for both cell lines. E.g. how does THBS1 knock down influence the invasive capacity or the total tumor area in NCH421K cells or P3 cells, respectively?

28. Discuss in more detail why THBS1 is differentially regulated by TGF β in NCH421K vs P3 cells in vitro and in vivo. How could THBS1 inhibition enhance the efficacy of Bevacicmab?

Minor concerns:

1. Suppl. Fig. 4: Please label the x and y axis appropriately.

2. Reference 24, in page 7, line 174 is not correct. Seliger et al. show a correlation between THBS1 and transforming growth factor 2 not 1.

3. Why do the headings of the figure legends of supplemental figure 8, 9 and 10 have a hyphen? Please be consistent throughout the text.

4. Please shortly explain all abbreviations in the text before using them the first time (e.g. "TCGA" in line 155). It is not sufficient to state their use only in the abbreviation section.

5. Suppl. Fig. 1C: Which gene cluster does this diagram refer to? It is neither described in the

figure nor in the figure legend.

6. The explanation regarding the Xenome analysis is insufficient. A more detailed description would be necessary to understand this part of the results properly (e.g. what does connectivity to a certain compartment mean?).

7. The number of patient samples and their characteristics should be described in the material and methods part.

Response to Reviewers' comments:

Reply to the overall main comment:

We had submitted this work previously to Nature communications and it went through the review process. Unfortunately, the article was not accepted by Nature communications at that time. We have since then completely reshaped this work and performed many new experiments. The article, in the present form represents an entirely new article and constitutes a very solid piece of work.

The following critical issues were adjusted in the manuscript when compared to the previous version:

1/ The manuscript is focused specifically on THBS1 where a systematic analysis has been conducted in many *in vitro*, *ex vivo* and *in vivo* models

2/ The manuscript studies primarily tumor cell-derived THBS1. The article focuses on the role of THBS1 in both GBM expansion and invasion which entails angiogenesis-dependent and independent mechanisms.

3/ The manuscript is based on the use of classical (U87 etc..) and PDX cell lines which were systematically compared in most of the experiments

4/ The infrared results were removed from the manuscript since they do not add any significant information to the understanding of the role of THBS1

5/ The SMAD reporter assays were conducted in cell lines AND in a PDX GBM model which further validates this mechanism

6/ We have added specific experiments with regard to tumor cell-bound CD47 using various assay that demonstrate a role in tumor cell invasion

7/ To simplify the message of the manuscript, we removed some of the data with regards to NCH421K cells. Since NCH421K cells are genetically very different to the patient-derived cell lines we used throughout the study.

8/ Several minor technical issues were also addressed

We believe that our results are of significance to further understand the role of this important extracellular protein in glioma biology.

Reviewers' comments:

Reviewer #1 (Remarks to the Author):

Overall criticism:

In this manuscript, Daubon et al. reported that THBS-1 highly expressed in the invasive edge and the tumor vessels promotes GBM tumor invasion. They identified the differential expression of THBS-1 by RNA-seq comparison between cells from invasive areas and angiogenic areas. They showed that TGF β is in the upstream of THBS-1 and induces THBS-1. Binding of the phosphor-SMAD2/3 to THBS-1 promoter may be responsible for the upregulation of THBS-1 by TGF β stimulation. They showed that THBS-1 may inhibit the RhoA inhibitor RND3, and thus activate RhoA to promote cell migration and invasion. In the orthotopic xenograft model, they showed that knock-down of THBS-1 inhibited tumor invasion, extended mouse survival, and suppressed tumor vascularization. Finally they used Bev, THBS-1/LAP inhibitor LSKL, and THBS-1/CD47 inhibitor TAX2 to test the effect of combined treatments on tumor invasion and vascularization. Although the topic of GBM invasion is interesting, the main conclusions of this manuscript are not supported by the current data. The significance of the manuscript is severely compromised by several critical concerns.

Reply:

We have now revised our manuscript completely and answered the concerns raised by the reviewers. We have in particular simplified the message of this manuscript, removed unnecessary data and added a number of new and critical experiments. This manuscript constitutes now an entirely new piece of work. We would like to thank the reviewer for his insightful comments.

Criticism 1:

The manuscript identified THBS-1 as an invasion-associated gene by comparison of micro-dissected samples from so-called invasive areas and angiogenic areas. How did the authors define the invasive areas and angiogenic areas? Did they specifically choose or avoid vessel cells that should exist in both invasive and angiogenic areas? Aside from the invasive areas and angiogenic areas, are the remaining regions of tumor mass analyzed? The authors should present a representative IHC image for the criteria they applied to get the cells for RNA-seq.

Reply 1:

Angiogenic areas are located in the viable tumour part near the tumour center, where significant angiogenesis is observed. Invasive areas are defined as located at the periphery of the tumour and these may be either collective strand invasion or single cell invasion. In our case, the samples used were extracted from the viable part of the tumour center and from the invasive strand, and these were used for RNA sequencing as shown in Figure 7 (**Figure 1 below**). In Figure 1C, the immunohistological picture of the orthotopically implanted tumour shown where the angiogenic and invasive areas (strand invasion) are clearly depicted.

Example of Laser-capture microdissected invasive area on P3 tumour (left is before LCM and right after LCM).

We took the whole tissue of both areas to perform RNA sequencing. It is evident that this will lead to both tumour-associated and stroma/vessel-associated genes. However and because of the nature of the tumour implantation (human tumour implanted in a mouse brain), and the small overlap of mouse and human genes (around 10%), we could easily separate the tumour cell-associated genes from the stroma- and vessel-associated genes (by using the Xenome program). A similar approach has been used in many of our papers where we implanted human tumour cells in the chicken chorioallantoic membrane (for e.g. Kilarski WW et al, 2018 or Soulet et al, 2013), the overlap in this case is only about 5%.

The remaining tumour regions consist of the necrotic core or tumour borders. We did not analyse the necrotic core for obvious reasons. Analysing the borders of the whole tumour would require the collection of many samples which goes beyond the scope of this article, since our intention was to compare an invasive area with the viable angiogenic core.

Criticism 2:

It seemed that TGFβ was randomly chosen for analysis of the THBS-1 signaling since they proposed that TGFβ may be an upstream factor of THBS-1 without any clue. Although they provided some evidence that TGFβ may activate the transcript of THBS-1 in P3 cells, their data also showed that TGFβ had no effect on THBS-1 in NCH421K cells. From these data, it is not convincing that TGFβ is a dominant regulator of THBS-1 in GBMs. The authors should test the expression of other reported upstream regulators of THBS-1.

Reply 2:

As stated in the response to the all criticisms, the structure of our article has been now entirely revised. We start from the hypothesis that THBS1 is critically involved in glioma development and invasion. We clearly show that TGFβ1 upregulates THBS1 expression but that THBS1 is not involved in TGFβ1 activation in the glioma context. We furthermore show convincingly using classical cell lines (U87) and patient-derived human cell lines that Smad3 is involved in the regulation of THBS1 by TGFβ1. Other regulators (EGF, PDGFbb, IL1β, etc, ...) did not significantly induce THBS1 (*Supplemental Figure 4C in the present manuscript* and below).

We furthermore show in two other patient-derived cell lines, TGFβ1 also upregulates THBS1 (*Supplemental Figure 4B in the present manuscript* and below).

Thus, to our opinion, these data show convincingly that TGFβ1 is an important regulator THBS1 in GBM. As stated to the reply to the overall comment, the data with regard to NCH421K were removed from the manuscript.

Criticism 3:

The source of THBS-1 is not clarified. Is it expressed by tumor cells? Or it is expressed by endothelial cells? If it is expressed by endothelial cells, do normal endothelial cells express THBS-1? Meanwhile, although the manuscript showed some evidence of the THBS-1 expression in tumor cells in invasive edge in P3 xenografts, they also showed universal expression of THBS-1 in NCH412K xenografts. Thus, it is possible that THBS-1 is universally expressed in tumor cells.

Reply 3:

THBS1 is expressed in both tumour-associated endothelial cells and tumour cells (as shown in **Figure 1B in the present manuscript** and below) and as published in recent paper by us (Bougnaud et al, Oncotarget 2016).

Endothelial and tumour cells analysed by sequencing = THBS1 was found as a central player of GBM development (Bougnaud et al, Oncotarget).

Expression is only found in tumour blood vessels but not in normal blood vessels (Figure above). In P3 tumours, the invasive strand clearly expressed increased amount of THBS1 as shown in **Figure 1C in the present manuscript**.

As stated to the reply to the overall comment, the data with regard to NCH421K were removed from the manuscript.

Criticism 4:

Whereas GBM invasion is surely a critical point for the malignancy of the tumor, due to the rapid tumor growth in mouse models, it is hard to directly link the tumor invasion to the animal survival. Given that the authors also showed that in the long time culture, knock-down of THBS-1 impaired tumor cell proliferation, it is likely that the tumor invasion is not an import issue for the delayed animal death after knock-down of THBS-1 with or without Bev treatment (Figure 5).

Reply 4:

THBS1 is clearly involved in both global tumour expansion and invasion. We have replicated our results in P3 tumours which are included in **Figure 3 of the present manuscript** and below. These results show that overall tumour growth, angiogenesis and tumour invasion are compromised after shRNA THBS1 expression in tumour cells. A fine analysis of blood vessel size distribution in shRNA treated tumours demonstrates small decrease in

small blood vessels (below 10 μm) but an increase in medium size blood vessels (between 10 and 20 μm). Thus, THBS1 depletion may alter the quality of the tumour blood vessels.

A

B

C

Criticism 5:

THBS-1 is reported as an inhibitor for angiogenesis by inhibiting VEGFR2 signaling. But the authors showed that shTHBS-1 dramatically inhibited tumor vascularization. They should interpret the phenomenon with some data.

Reply 5:

This criticism has been answered under reply 4.

Criticism 6:

The infrared photograph showed very different background signal in normal brain tissues in S.Fig3. This may be due to the position of the sections. However, this means that the infrared method is not very reliable.

Reply 6:

Infrared imaging data were removed because unnecessary for conveying the central message of our study.

Criticism 7:

How did the authors calculate the fold G4 vs G2 in Fig. 2A? The same table showed that the G2 mean is 6.32 and the G4 mean is 8.59.

Reply 7:

We apologized for the lack of explanation; these results were expressed in log2. This was modified in the table present in supplemental *supplemental Figure 1A* and below, and edited in the legend of the figure. These values are expression values in Log2 scale. This gives a log2 fold change of 8.59 minus 6.32 = 2.27. To convert this to a linear fold change we calculate 2 to the power of 2.27 ($2^{2.27} = 4.82$).

GENE_SYMBOL	fold G4 vs G2 (log2)	G2 mean	G3 mean	G4 mean	Absolute t value (G2 vs G4)	Raw p value (G2 vs G4)	Adj p value (G2 vs G4)
THBS1	4.82	6.32	6.77	8.59	5.77	1.69E-07	2.39E-06
TGFBI	1.53	9.13	9.39	9.75	4.21	6.12E-05	4.05E-04

Criticism 8:

Please enlarge the Fig 2C for grade II and III sections, so that the cell types with THBS-1 staining could be determined.

Reply 8:

As the reviewer requested, we show now a new enlarged figure (*Figure 1B of the present manuscript referring to old figure 2C* and below). In the new figure, we can easily distinguished THBS1 expression in tumour cells in grade II and grade III glioma (highly expressed in endothelial cell in grade III glioma).

Criticism 9:

In Fig. 2B, although the 2 grade IV tumor samples had relatively high THBS-1 expression, the difference between these 2 samples was dramatic.

Reply 9:

We have analysed 12 patient samples from each grade and quantified the intensity of THBS1 signal. We now show a new figure that is more representative of the quantification of THBS1' signal intensity (*Figure 1A of the present manuscript* - and below).

Criticism 10:

For all immunoblot showing p-SMAD2/3, the total SMAD2 and SMAD3 should be shown.

Reply 10:

This was modified in the *supplemental figure 4A of the present manuscript* and below.

Criticism 11:

Fig. 3D showed no significant difference in the immunoblot.

Reply 11:

The experiment was repeated and the data reanalysed which is depicted now in the *supplemental Figure 5B* and below.

Criticism 12:

Fig. 3F showed that deletion of SMAD3 binding site#1 significantly increased the activation of THBS-1 promoter even without TGFβ stimulation. Please explain.

Reply 12:

To corroborate these results, we have performed additional experiments using a patient-derived cell line. These results confirmed that the deletion of Smad3 binding site increase the activation of THBS1 promoter without any TGFβ activation. Smad3 binding site may function as a repressor and the net result of THBS1 activation by TGFβ may be due to the ratio of Smad binding to binding site 1 and 2 respectively (*Figure 2E of the present manuscript* and below). This is now discussed in the manuscript in the discussion part.

Criticism 13:

Fig 4D and 4E showed wound healing, which does not represent cell invasion but only cell migration. So the statement in the text was wrong.

Reply 13:

In this assay, the tumour cells were included between two layers of Matrigel and thus must move through an organized matrix. This is clearly a hallmark of invasion and thus the assay represents an invasion assay.

Criticism 14:

In Fig.5, the authors regarded Bev treated xenografts as hypoxic samples. They should provide some direct evidence that the samples did contain more hypoxic regions and the THBS-1 expression was in these regions.

Reply 14:

For patient-derived tumour cells implanted in the mouse or rat brain undergo hypoxia after bevacizumab treatment as shown in two of our articles (Keunen et al PNAS 2011, Fack et al Acta Neuropath 2015, and below).

Rat brain with FMISO in PET-SCAN, higher signal after bev treatment (right) – Fack et al Acta Neuropath 2015

We also analysed hypoxia in our tumours by using an indirect marker, the Carboxic Anhydrase IX (CAIX). CAIX, which catalyses the interconversion of carbon dioxide and water to bicarbonate and protons, has been shown to be regulated by hypoxia. We show in a patient sample that a strong staining for CAIX in peri-necrotic regions which indicated high hypoxia (*Supplemental figure 2C* and below). Furthermore in these regions, THBS1 was co-expressed with CAIX.

Criticism 15:

It is strange that all the invasions the manuscript showed looked like protrusions. Did they observe any scattering spots as the consequence of invasion?

Reply 15:

As stated in the manuscript, collective strand invasion was analysed in detailed in our *in vivo* experiments. As requested by the reviewer, we have now added quantifications of single cell invasion to our results (**Figure 5C of the present manuscript** and below).

Criticism 16:

What does vascular coverage in Figs. 6 and 7 mean? Do they mean vessel density? How did they calculate it?

Reply 16:

We apologize for this misunderstanding; we replaced vascular coverage by vascular density, which was calculated as the ratio of CD31 positive vessel staining and Nestin staining of the whole tumour area (this was edited in the text).

Criticism 17:

No vessels can be seen in Fig. 6F.

Reply 17:

As stated to the reply to the overall comment, the data with regard to NCH421K were removed from the manuscript.

Reviewer #2 (Remarks to the Author):

Overall criticism:

Daubon et al. claim that TGF β 1-induced expression of THBS1 increases invasion in murine glioblastoma models. Furthermore, they demonstrate that silencing of THBS1 has an additive therapeutic effect with Bev in an *in vivo* GBM model. It is although troublesome that the shRNA used *in vivo* did not induce efficient knockdown *in vitro*. While the findings are of potential interest, the data provided do not convincingly support the claims of the authors. Moreover, the correlation between THBS1 and THG β is not novel (Seliger et al., 2013).

Although they provide mechanistic *in vitro* data on the induction of THBS1 by TGF β 1 in U87 cells, there are major concerns regarding their validation *in vivo* as well as other aspects of the study. As the data differ significantly between the P3 and NCH421K model and also from other published data in other entities (e.g. Fernando et al., Clinical Cancer Research 2008), their implication and reproducibility is unclear. Additional *in vivo* models and larger patient cohorts showing correlation of THBS1 level and meaningful clinical parameters with sound statistics are necessary to support the main claim of the manuscript. Furthermore, Gain-of-Function approaches with overexpression of THBS1 *in vitro* and *in vivo* are desirable to proof the underlying mechanism.

Reply:

We have carefully read the paper by Seliger et al and compared this to the data we are providing in our manuscript. In their manuscript published in Plos One in 2013, the authors report that lactate increases THBS1 expression which in turn contributes to TGF β 2 activation. Our data are completely opposed to these published results because we show using cell lines and patient-derived models (P3, BL13 and BL9) that TGF β 1 is not activated by THBS1 but, on the contrary, TGF β 1 induces THBS1 expression. Furthermore, we show for the first time in a standard and in patient-derived cell line that the mechanism involves a Smad3-dependent transcriptional regulation. Thus, our data are highly novel and of significance for the THBS1/TGF β 1 interaction.

We have now performed many additional experiments that fully support the claims made in our manuscript:

- 1/ The manuscript is focused specifically on THBS1 where a systematic analysis has been conducted in many *in vitro*, *ex vivo* and *in vivo* models;
- 2/ The manuscript studies primarily tumor cell-derived THBS1. The article focuses on the role of THBS1 in both GBM expansion and invasion which entails angiogenesis-dependent and independent mechanisms;
- 3/ The manuscript is based on the use of classical (U87 etc..) and a PDX cell line which were systematically compared in most of the experiments;
- 4/ The infrared results were removed from the manuscript since they do not add any significant information to the understanding of the role of THBS1;
- 5/ The SMAD reporter assays were conducted in cell lines AND in a PDX GBM model which further validates this mechanism;
- 6/ We have added specific experiments with regard to tumor cell-bound CD47 using various assay that demonstrate a role in tumor cell invasion;
- 7/ To simplify the message of the manuscript, we removed data with regards to NCH421K cells, because NCH421K are of proneural phenotype. Since NCH421K cells are genetically very different to the cell lines we used throughout the study (P3 and U87 with mesenchymal characteristics).
- 8/ Several minor technical issues were also addressed.

We have now changed entirely the order of the manuscript. The manuscript focuses primarily from the beginning on the various functions of THBS1 in GBM development which were analysed in a great number of experiments. The RNA sequencing data were put as supporting data at the end of the manuscript which indeed showed that THBS1 is among the two genes with the highest connectivity. This result from an unbiased approach greatly supports the contentions of the manuscript.

We have conducted many additional experiments including *in vivo* studies (survival studies) which in our hands provides enough results to support the contentions of our study. In particular, we performed many experiments by using classical cell lines and also patient-derived glioma model. The clinical data we used are from Sun et al (Cancer Cell 2006) and TCGA. The Sun data showed linear increase in THBS1 expression with grade (45 Grade

II patients, 31 grade III patients and 81 Grade IV patients) and analysis by TCGA showed significant correlation to survival (bad prognosis). The western-blot in **Figure 1A** of the present manuscript is supporting these large scale data.

We did not overexpress THBS1 in glioma cells because to our opinion this will not add anything to reinforce the message of the paper. However, as the reviewer suggested, we have already prepared lentiviral constructions for overexpressing THBS1 in patient-derived cells. We have not yet generated functional data with this construct but we are ready to do so in the near future.

The aim of our manuscript was to address the role of endogenous THBS1 in glioma development and invasion. This is very different from the data obtained by Fernandez et al in Clin Can Research in 2008 because these authors studied the effect of exogenous THBS1 that was transfected into colon and kidney cancer cells, which led to very high expression levels. Thus, the authors did not only NOT investigate endogenous THBS1 in their tumor models but also used tumor cell lines that are very different from glioblastoma.

Furthermore, our results are in agreement with the data published in 2011 in Cancer Research (Firlej V. et al) where a pro-tumor and pro-invasive effect of THBS1 was evidenced, albeit using another tumour model (prostate cancer).

Criticism 1:

Authors should rewrite the title because it appears overstated. From their results they cannot conclude that thrombospondin-1 contributes to TGF β 1-dependent invasion in glioblastoma.

Interaction between TGF β 1 and THBS1 is only shown in U87 cells in vitro. More cell lines and in vivo data from different GBM models are necessary to validate this mechanism.

Reply 1:

According to the suggestions to the reviewer, we modified the title of the manuscript. The title is now “Deciphering the complex role of thrombospondin-1 in glioblastoma development”.

As already explained in the overall reply to the reviewer, we added a great number of new experiments. In particular, we validated the effect of THBS1 by TGF β 1 in three patient-derived cell lines and studied the precise transcription regulation in the P3 patient-derived cell line (**Figures 2 and 3 from the present manuscript**). These data are in agreement with what has been found in the U87 cell line.

Criticism 2:

It is unclear how authors come to the conclusion that THBS1 is 4.8-fold upregulated in G4 vs. G2 when the mean is 8.6- and 6.3-fold in G4 vs. G2, respectively. This needs to be explained better.

Reply 2:

We apologized for the lack of explanation, these results were expressed in log₂. This was modified in the table present in **supplemental Figure 1A in the present manuscript**, and edited in the Material and Methods part. These values are expression values in Log₂ scale. This gives a log₂ fold change of 8.59 minus 6.32 = 2.27. To convert this to a linear fold change we calculate 2 to the power of 2.27 ($2^{2.27} = 4.82$).

Criticism 3:

Page 6, line 134: The authors state that the P3 and the NCH421K tumor model show different results regarding the expression of THBS1 due to heterogeneity of glioblastoma. Checking two tumor models is not sufficient to support this statement. The authors need to investigate more models. Currently, the results from one model cannot be validated in the other, thus their relevance is unclear.

Reply 3:

As explained in the overall criticism, we completely reorganized the manuscript and added a number of additional data. The manuscript focuses primarily from the beginning on the various functions of THBS1 in GBM development which were analysed in a great number of experiments using classical cell line (U87) and patient-derived tumour models (P3 and BL9/BL13). The RNA sequencing data were put as supporting data at the end of the manuscript which indeed showed that THBS1 is among the two genes with the highest connectivity. This result from an unbiased approach greatly supports the contentions of the manuscript. Furthermore, as the reviewer requested we have performed expression analysis using additional patient-derived tumour models (BL9 and BL13) and shown that THBS1 has increased expression in peripheral tumour areas (see below).

As explained in the overall criticism of the manuscript, the data from the NCH421K cells have been removed and since P3 and NCH421K are genetically different.

Criticism 4:

It is not clear which conclusions are drawn from the IR microscopy and how these analyzes relate to THBS1 and the idea of the paper. This is further complicated by the fact that results are not consistent among both tumor models.

Reply 4:

To simplify the message of the manuscript, we have removed the infrared microscopy data since they will not add anything new which would support the contentions of the manuscript.

Criticism 5:

Figure 2C: The authors should provide histomorphometric quantification and statistical tests instead of representative pictures to quantify THBS1 expression in tumors. The number of analyzed patients should be stated in the figure legend as it was done for figure 2B.

Reply 5:

The histological figures are only there to illustrate in a qualitative manner the expression of THBS1 in patients. This figure is now in *supplemental Figure 2A*. However to satisfy the reviewer, we performed quantifications of immunostainings from 6 different patients. The molecular profiles of the different tumours are indicated in the table below. These results are now included in *Supplemental Figure 2B* and also shown below.

	Molecular tumour profile
Patient 1	GBM IDH1 wt MGMT + (59%) MIB1:45%
Patient 2	GBM MIB18% IDH1 wt MGMT met
Patient 3	GBM IDH1wt MGMT met (42%) MIB1 10%
Patient 4	GBM IDH1 wt MGMT met (35%) MIB1 10%
Patient 5	GBM IDH1 wt MIB1 15% MGMT neg 1p+ 19q-
Patient 6	GBM IDH1 wt MGMT met (8%) MIB1 20%

Criticism 6:

The quality of the figures needs significant improvement throughout the manuscript, this applies especially to the immunohistochemistry pictures. Here, also higher magnifications need to be displayed. The differences described in the manuscript for example in figure 2D and E are not evident from the pictures. Also it appears rather uncommon to label Figures as e.g. 2Db.

Reply 6:

According to the reviewer suggestions, we have improved significantly the quality of the IHC pictures and added magnifications (*Figure 1B of the present manuscript*).

Criticism 7:

The authors claim that THBS1 expression is highest in peri-necrotic and hypoxic areas but hypoxic areas cannot be characterized by morphological criteria. Tumor border areas were chosen to measure THBS1 signal intensities. Those tumor areas are usually rather well oxygenated instead of being hypoxic. If the authors wish to prove that THBS1 expression predominantly occurs in hypoxic tumor areas, murine tumor models are warranted including injections of the hypoxia marker pimonidazole.

Reply 7:

Instead of using pimonidazole, we used an indirect readout for hypoxia which is Carbonic Anhydrase IX (CAIX), commonly used by neuropathologists. CAIX, which catalyses the interconversion of carbon dioxide and water to bicarbonate and protons, has been shown to be regulated by hypoxia (Allen E. et al, Sci Transl Med. 2017). These stainings are shown in the figure below (*Supplemental Figure 2B of the manuscript* and below). In the central tumour area, there is a strong induction of hypoxia and THBS1. In the tumour margins, only few cells stained positively for CAIX, and there is no co-expression of CAIX and THBS1.

This is due to the fact (and as the reviewer correctly pointed out) that tumour borders are usually well oxygenated. We assume that tumour cells from hypoxic areas are migrating to tumour borders and lose progressively CAIX expression, albeit maintaining THBS1 expression. This is reinforced by recent experiments

we conducted using fetal brain aggregates with P3 tumor spheroids transduced with rainbow vector. Tumor cells are highly motile and are able to migrate from the center to the periphery or from one end of the spheroid to the other by migrating through the entire spheroid.

Criticism 8:

Figure 2 E: The authors claim higher THBS1 levels in invasive areas using IHC. It is not clear how authors discriminate between angiogenic and invasive areas as this is not evident from the pictures provided.

Reply 8:

Angiogenic areas are located in the viable tumour part near the tumour center, where significant angiogenesis is observed. Invasive areas are defined as located at the periphery of the tumour and these may be either collective strand invasion or single cell invasion. In our case, the samples used were extracted from the viable part of the tumour center and from the invasive strand, and these were used for RNA sequencing as shown in Figure 7. In *Figure 1C* (below), the immunohistological picture of the orthotopically implanted tumour shown where the angiogenic and invasive areas (strand invasion) are clearly depicted.

Criticism 9:

Figure 2E: It is of concern that the data regarding THBS1 expression is not consistent among both tumor models.

Reply 9:

We already replied to this comment in **reply 3**.

Criticism 10:

It is difficult to understand, why the authors use the cell line U87 in Figure 3, 4 and 5 to study the mechanisms, pathways and biological effects of THBS1 and TGF β 1 while other experiments are performed with P3 or NCH421K cells. Cell lines and models need to be used more consistently in vitro and in vivo.

Reply 10:

The manuscript is based on the use of classical (U87) and PDX cell lines (P3, BL9 ad BL13) which were systematically compared in most of the experiments. We have now changed the entire flow of the manuscript by systematically comparing the U87 cell line and the P3 patient-derived cells. The data from the P3 and the U87 models are overall in agreement. For example, a critical aspect was the mechanistical study with regard to the regulation of THBS1 by TGF β . Indeed identical results, involving the Smad3 as a critical regulatory loop has been both validated in the P3 and the U87 models.

Criticism 11:

Figure 5A: How was the quantification of THBS1 intensities done - no description is provided in the methods section. Detailed description of how statistical significance was evaluated should be provided.

Reply 11:

The quantification was done by measuring the intensity of THBS1 staining as grey values and comparing this to the tumour area in respective fields (determined by Vimentin stainings). These measurements were done on 5 different tumours and 2 sections *per* tumour. However, these data were removed from the manuscript and the figure moved to the supplemental data section because they represent supporting information only.

Criticism 12:

Figure 5F: How can this strong, synergistic effect of Beva + shTHBS1 be explained in relation to the rather small anti-tumor effect of the single treatments? How do the NCH421K and P3 PDX models respond to the combinatorial treatment?

Reply 12:

Bevacizumab treatment alone and shTHBS1-2 treatment alone led to an equivalent increase of life span of animals implanted with U87 cells. Bevacizumab treated shTHBS1-2 tumours exhibited a strong survival increase because of the additive/synergistic effect. This may be due to the blockade of bevacizumab-dependent effects on THBS1 expression.

As the reviewer suggested we performed the combinatory treatment on P3 tumours. This experiment is still on going. Preliminary results indicate that combinatory use of bevacizumab and shTHBS1 prolongs survival (see below).

In addition, we performed a new experiment with P3 cells with two different shTHBS1 which demonstrated an increase in survival as shown below (*Figure 3C in the present manuscript*).

Criticism 13:

Suppl. Fig. 6B: The staining for THBS1 is hardly visible in any of both pictures. Please replace them.

Reply 13:

This figure is now significantly improved as shown below (*Supplemental Figure 4D of the manuscript*).

Criticism 14:

Figure 6A and 6D, why does P-SMAD have one band in P3 cells and two bands in NCH421K cells?

Reply 14:

We have replaced the figure with additional western-blot for total and phosphorylated Smad2 and Smad3 antibody recognizing both total and phosphorylated forms (*Supplemental Figure 4A and below*). The data from the NCH421K were removed from the manuscript.

Criticism 15:

Suppl. Fig. 9A: The western blot for THBS1 is of poor quality. The bands cannot be appreciated properly. Please replace them.

Reply 15: T

The western blot was replaced in the *Supplemental Figure 7A* (and below).

Criticism 16:

The authors intend to show that THBS1 exerts its function on glioblastoma cells by decreasing RND3 level which subsequently activates RHOA. So far, the data are correlative and descriptive. Authors should provide mechanistic data by Gain- and Loss-of-function experiments.

Reply 16:

These data were removed because they do not add substantial information for the contentions of this manuscript.

Criticism 17:

Page 9, line 217 and Figure 4A: Authors state that THBS1 expression was strongly reduced with both shRNAs”, however, there is only a marginal reduction of THBS1 expression using shTHBS1-1 (reduction from control is only 16 %). It is therefore hard to understand why shRHBS1-1 and shTHBS1-2 have the same effect on RND3 expression and wound healing when compared to the shRNA control. This suggests that the observed effects might not be specific.

Reply 17:

The small reduction of THBS1 signal in the western-blot was due to loading differences. This was now corrected and we have thus replaced the western-blot image a new western-blot made from the same cell extracts and the image is shown below (*Supplemental Figure 6A in the manuscript*). Downregulation of THBS1 when compared to control is highly significant in both shTHBS1 treated cells.

Criticism 18:

As shTHBS1-1 does not seem to reduce THBS1 expression, it is important that authors specify, which shRNA has been used for THBS1 knock down in the in vivo experiments (Figure 6). Please refine the figure legends accordingly.

Reply 18:

We already addressed this issue in our **reply 17**. The graph depicts the survival curves of animals transplanted with shTHBS1-1-transduced U87 cells (*Supplemental Figure 8C in the present manuscript* and below).

Criticism 19:

U87 cells do not seem to express high levels of THBS1. In addition to shRNA mediated knock down it would therefore be necessary to also overexpress it in U87 cells and study the biologic effect *in vitro* and *in vivo*.

Reply 19:

We have generated a clone with very high THBS1 expression in U87 cells, U87 IRE1dn cells (*Supplemental Figure 3 in the present manuscript* and below). These cells exhibit very high THBS1 expression and have a highly invasive phenotype *in vivo* (expressed 200-300 pg THBS1 /10⁶ cells). The biological effects of these cells has been extensively studied by us in the previous publications (for e.g. Auf et al PNAS, 2011) and in the present manuscript.

However, as the reviewer suggested, we have already prepared lentiviral constructions for overexpressing THBS1 in patient-derived cells. We have not yet generated functional data with this construct but we are ready to do so in the near future.

We think that for the message conveyed by this manuscript, the present data are, to our opinion, sufficient.

Criticism 20:

Suppl. Fig. 7: It is not described in the text or in the figure legend in which cells the experiment has been performed.

Reply 20:

These data were removed because unnecessary for the message conveyed by the manuscript.

Criticism 21:

Suppl. Fig. 8B and C: The images are blurry. Please replace them.

Reply 21:

These data were removed because unnecessary for the message conveyed by the manuscript.

Criticism 22:

It is not sufficient to perform shRNA-mediated knock down experiments with only one shRNA. The in vivo and in vitro experiments need to be repeated with another shRNA against THBS1 to rule out off-target effects.

Reply 22:

We have repeated the experiments with more than one shRNA as shown below (*Figure 3 of the present manuscript*).

Criticism 23:

THBS1 has been well-studied and characterized as a potent anti-angiogenic mediator. It is therefore difficult to understand that its absence does not change (Figure 6C) or even decrease (Figure 6 F) vascular coverage of glioblastoma in vivo. This is also not in line with previous data showing re-expression of endoglin upon THBS1 knock down.

Reply 23:

THBS1 is clearly involved in both global tumour expansion and invasion. We have replicated our results in P3 tumours which are included in *Figure 3* in the present manuscript. These results show that overall tumour growth, angiogenesis and tumour invasion are compromised after both shRNA THBS1 expression in tumour cells. A fine analysis of blood vessel size distribution in shRNA treated tumours demonstrates small decrease in small blood vessels (below 10 μm) but an increase in medium size blood vessels (between 10 and 20 μm). Thus, THBS1 depletion may alter the quality of the tumour blood vessels.

Criticism 24:

Bevacizumab is widely used as a therapy in murine glioblastoma models. It is therefore surprising that it does not reduce the tumor load although it was used at common dose levels (Figure 7B and C). This needs to be explained.

Reply 24:

The P3 model when grown in standard culture conditions (Agar layer and DMEM) is not decreasing tumour load after bevacizumab treatment (Keunen et al, PNAS 2011). In old Figure 7B, cells cultured under these conditions were implanted in mice and tumours were analysed at day 35 after surgery. In these experiments, an effect of TAX2/Bev treatment on tumour cell invasion was observed. However, when cells are grown in neurobasal medium, tumours become sensitive to bevacizumab treatment as evidenced by an increase in survival of tumour-bearing mice (**Figure 5D in the present manuscript** and below).

Criticism 25:

The authors showed that loss of THBS1 signaling reduces invasion thereby prolonging survival of tumor-bearing mice (Figure 5). It is of concern that THBS1/CD47 blockade with TAX2 does not reduce invasion in both tumor models. This is of special importance as TAX2 treatment does not seem to have an effect on tumor load. Please explain.

Reply 25:

We have performed now *in vitro* experiments on U87 and P3 cells where CD47 has been knocked-down. U87 cells show a reduction in cell migration and P3 cells inhibition of cell invasion (**Figure 6 in the present manuscript and below**).

When the same experiment was carried out with TAX2, inhibition was observed in hypoxic conditions which we explain by hypoxia-induced THBS1 expression (**Figure 5A of the present manuscript**, and below).

In addition to strand invasion, we now analysed our tumours for single-cell invasion. We observed that single-cell invasion was inhibited to some extent *in vivo* in the P3 model by TAX2 alone and in combination with bevacizumab, whereas collective strand invasion was only inhibited by the TAX2 and bevacizumab combination. We assume that tumour cells from hypoxic areas are migrating to tumour borders, albeit maintaining THBS1 expression.

Criticism 26:

If the authors want to show that inhibiting THBS1/CD47-signaling holds potential as a therapeutic approach, please provide survival data for tumor bearing-mice treated with Beva +/-TAX2 (e.g. for P3 and NCH421K from Figure 7).

Reply 26:

We performed a survival experiment on P3 tumours, as shown below. When TAX2 was added to Bevacizumab, survival was increased (50% of survival for Bev alone is 62 days and 50% of survival for Bev and TAX2 combination is 68 days).

**Criticism 27:**

Figure 7: Consistent readouts should be provided for both cell lines. E.g. how does THBS1 knock down influence the invasive capacity or the total tumor area in NCH421K cells or P3 cells, respectively?

Reply 27:

Knock down of THBS1 in P3 tumour leads to significant increase in survival (two different shRNAs were used in this experiment; **Figure 3C in the manuscript** and **Reply 22**). Contro-lateral invasion was also inhibited for cells transduced with both THBS1 shRNAs (**Figure 3B in the manuscript** and **Reply 22**).

As stated to the reply to the overall comment, the data with regard to NCH421K were removed from the manuscript.

Criticism 28:

Discuss in more detail why THBS1 is differentially regulated by TGFβ in NCH421K vs P3 cells in vitro and in vivo. How could THBS1 inhibition enhance the efficacy of Bevacicumab?

Reply 28:

The manuscript is now completely reorganized and we have systematically compared the U87 cells and the patient derived cell model P3. We evidenced the same regulatory mechanisms for both cell types. For this reason, we have removed the NCH421K data from the manuscript.

Bevacizumab treated shTHBS1-1 tumours exhibited a strong survival increase because of the additive/synergistic effect. This may be due to the blockade of bevacizumab-dependent effects on THBS1 expression. An alternative explanation may be that THBS1 blockade impairs evasive resistance induced by bevacizumab treatment by limiting the pro-invasive effects (Lu K et al, Cancer Cell 2012).

Criticism 29:

Suppl. Fig. 4: Please label the x and y axis appropriately.

Reply 29:

Appropriate labelling has now been done in the new Suppl. Figure 3C.

Criticism 30:

Reference 24, in page 7, line 174 is not correct. Seliger et al. show a correlation between THBS1 and transforming growth factor 2 not 1.

Reply 30:

The text was modified and a discussion of the article by Seliger et al is now included (Discussion section in the present manuscript).

Criticism 31:

Why do the headings of the figure legends of supplemental figure 8, 9 and 10 have a hyphen? Please be consistent throughout the text.

Reply 31:

This was corrected.

Criticism 32:

Please shortly explain all abbreviations in the text before using them the first time (e.g. “TCGA” in line 155). It is not sufficient to state their use only in the abbreviation section.

Reply 32:

This was corrected.

Criticism 33: Suppl. Fig. 1C: Which gene cluster does this diagram refer to? It is neither described in the figure nor in the figure legend.

Reply 33: The figure depicts the interaction network of TGFβ1 with genes expressed preferentially in angiogenic or invasive areas. Arrows represent potential activation or inhibition by TGFβ1 of these targets. The legend was modified and replaced in the present manuscript by :

“Interaction network (from Ingenuity Pathway Analysis) showing known gene-gene relationships involving TGFβ1 and genes differentially expressed between the angiogenic and invasive areas in P3 tumour. The ratio of expression of genes in the angiogenic and invasive areas has been overlaid on to the network. Genes expressed higher in the angiogenic areas are green while genes expressed higher in invasive areas are pink. Arrows represent potential activation or inhibition by TGFβ1 of these targets, including THBS1.”

Criticism 34: The explanation regarding the Xenome analysis is insufficient. A more detailed description would be necessary to understand this part of the results properly (e.g. what does connectivity to a certain compartment mean?).

Reply 34: The text was modified *in the present manuscript* : “Xenome effectively reconstructs the transcriptomes of multiple species from mixed RNA samples”.

And “TNF and TGFβ1 represent a class of powerful growth factors with important roles in tumour development. To examine further the role of secreted factors in communication between stroma and tumour, we constructed putative protein-protein interaction networks integrating the tumour and stroma transcriptomes (Figure 7). These networks represent soluble factors and receptors that were either over-expressed in tumour cells from infiltrative areas or expressed at significant levels in the surrounding stroma (see methods). Thrombospondin-1 (THBS1), Annexin II (ANXA2), and PDGFB were found to have the highest connectivity in the resulting networks (Fig 7A-B and Table S5). In other words, these molecules represent factors expressed in infiltrative tumour areas that have many known interactions with molecules that we have shown are expressed in the surrounding stroma.”

Criticism 35:

The number of patient samples and their characteristics should be described in the material and methods part.

Reply 35:

The clinical data we used are mainly from TCGA which show a significant increase in THBS1 expression and a correlation to survival. The western-blot we show in Figure 1A is supporting these large scale data (12 samples of patients were analysed for each grade).

For the immunostainings, a representative sample from one patient is shown –patient 1 in the table below (see **reply 5**). Furthermore we have added a figure that depicts the quantification of THBS1 expression in 6 patients (below).

	Molecular tumour profile
Patient 1	GBM IDH1 wt MGMT + (59%) MIB1:45%
Patient 2	GBM MIB18% IDH1 wt MGMT met
Patient 3	GBM IDH1wt MGMT met (42%) MIB1 10%
Patient 4	GBM IDH1 wt MGMT met (35%) MIB1 10%
Patient 5	GBM IDH1 wt MIB1 15% MGMT neg 1p+ 19q-
Patient 6	GBM IDH1 wt MGMT met (8%) MIB1 20%

Reviewers' Comments:

Reviewer #1:

Remarks to the Author:

The revised manuscript by Daubon et al. was almost thoroughly re-organized and could be regarded as a new manuscript. In this new version, the authors described the upregulation of THBS-1 in high grade gliomas. From a past observation of the simultaneous upregulation of TGF-beta and THBS-1 in the IRE-1 knock-down GBM cells, they started to investigate the connection between TGF-beta and THBS-1. They found that THBS-1 was upregulated by TGF-beta through TGF-beta-induced activation of Smad-3. In cell and xenograft models, they found that THBS-1 promotes tumor cell invasion. The authors showed the inhibition of invasion by either disruption of THBS-1 or treatment with the TGF-beta inhibitor. They also showed a link between THBS-1 expression and hypoxia, which is a consequence of anti-VEGFR treatment. Finally, they showed that THBS-1 may bind to CD47 in tumor cells to induce invasion.

The revised manuscript is significantly optimized from many aspects. The mechanisms underlying THBS-1 induction were investigated in depth. The possible clinical significance of THBS-1 in GBM treatment was also clearly demonstrated. The authors may still want to clarify the following issues.

1. The authors kept in using the terms "angiogenic (core) area" vs. "invasive area", which is quite confusing. As shown in Fig.1 and other figures, the invasive area still has many CD31+ vessels and thus is still angiogenic. The authors may want to rename the "angiogenic area" as "core area" or "main tumor body".
2. Whereas the authors showed that THBS-1 is expressed in both tumor and endothelial cells, they stated that THBS-1 is mainly expressed in tumor cells in some areas and in endothelial cells in other areas. It is hard to see such difference in the figures. Thus, those incorrect statements can be removed.
3. The data related to CD47 were not solid. The authors may want to use THBS-1 to treat shCD47-expressing cells to demonstrate that CD47 mediates the response of tumors to THBS-1 stimulation.
4. Some data were not representative. For example, in Fig. 1A, the tubulin in samples from GradeIII and GradeIV tumors showed a dramatic difference, which may lead to the difference of THBS-1 levels between the tumors. In addition, whereas Fig 1A showed GradeIII has more THBS-1 than GradeII tumors, Fig.1B showed the opposite staining.
5. When introducing the reasons for the investigation of the link between THBS-1 and TGF-beta, the authors may want to make a clear statement of the observed association between the two factors in the IRE-1 knock-down cells.
6. Given that the authors described the upregulated transcription of THBS-1 by TGF-beta, they should show some qPCR data to demonstrate the upregulated mRNA levels in core vs. invasive tumor areas.
7. In Page 9, Paragraph 1, the Fig 6C should be 5C, and the Fig 6D should be 5D.

Reviewer #2:

Remarks to the Author:

The data presented in the manuscript by Daubon et al. are not sufficiently supporting the conclusions of the authors, which compromises the significance of the manuscript.

Major comments:

The presentation of data is incomplete or insufficient at many places, for instance in the legend to Figure 1 it is claimed that n=12 different patient samples were analyzed for each grade. It is unclear why only two WB are shown, in addition the increase of THBS1 is accompanied by increases in tubulin when comparing Grade II with Grade IV. The IHC images are blurry and differences do not become obvious in Figure 1B. TMA of larger patient numbers need to be properly quantified. Similar comments apply at many other datasets when either data are missing

or not supporting the statements made by the authors.

Several of the previous concerns remain unanswered, for instance the Gain-Of-Function experiment and the combinatorial treatment experiment are still missing. It is not sufficient to state that experiments are ongoing.

REPLY TO THE REVIEWERS COMMENTS

MS: "Deciphering the complex role of thrombospondin-1 in glioblastoma development"

Reviewer #1 (Remarks to the Author):

The revised manuscript by Daubon et al. was almost thoroughly re-organized and could be regarded as a new manuscript. In this new version, the authors described the upregulation of THBS-1 in high grade gliomas. From a past observation of the simultaneous upregulation of TGF-beta and THBS-1 in the IRE-1 knock-down GBM cells, they started to investigate the connection between TGF-beta and THBS-1. They found that THBS-1 was upregulated by TGF-beta through TGF-beta-induced activation of Smad-3. In cell and xenograft models, they found that THBS-1 promotes tumor cell invasion. The authors showed the inhibition of invasion by either disruption of THBS-1 or treatment with the TGF-beta inhibitor. They also showed a link between THBS-1 expression and hypoxia, which is a consequence of anti-VEGFR treatment. Finally, they showed that THBS-1 may bind to CD47 in tumor cells to induce invasion.

The revised manuscript is significantly optimized from many aspects. The mechanisms underlying THBS-1 induction were investigated in depth. The possible clinical significance of THBS-1 in GBM treatment was also clearly demonstrated. The authors may still want to clarify the following issues.

1. The authors kept in using the terms "angiogenic (core) area" vs. "invasive area", which is quite confusing. As shown in Fig.1 and other figures, the invasive area still has many CD31+ vessels and thus is still angiogenic. The authors may want to rename the "angiogenic area" as "core area" or "main tumor body".

Reply: According to the reviewer suggestion, we have modified angiogenic areas and renamed them core areas.

2. Whereas the authors showed that THBS-1 is expressed in both tumor and endothelial cells, they stated that THBS-1 is mainly expressed in tumor cells in some areas and in endothelial cells in other areas. It is hard to see such difference in the figures. Thus, those incorrect statements can be removed.

Reply: According to the reviewer suggestion, we have removed the statements concerning the tumor/endothelial cell expression of THBS-1.

3. The data related to CD47 were not solid. The authors may want to use THBS-1 to treat shCD47-expressing cells to demonstrate that CD47 mediates the response of tumors to THBS-1 stimulation.

Reply: As the reviewer requested, we have now provided additional results to reinforce the contention that tumour cell bound-CD47 is involved in THBS-1-mediated effects related to tumor cell invasion. We have carried out an invasion assay using P3 cells that were transduced with shRNA control or shRNA CD47 lentiviral vectors and done the assay in presence or absence of purified thrombospondin-1 protein. The results clearly indicate that knock-down of CD47 impairs THBS1-induced tumor cell invasion. The results are presented in Figure 7C of the new manuscript and below.

We furthermore provided important additional *in vivo* results related to role of tumour-cell bound CD47 in glioblastoma development. P3 patient-derived CD47 knock-down tumour cells and control tumours cells were intracranially implanted in $Rag\gamma 2C^{-/-}$ mice, and tumour development was followed over 100 days. As shown in figure XX, survival was significantly increased in mice implanted with CD47 knock-down cells in comparison to control (***) $P < 0.001$). The results are presented in Figure 7D of the new manuscript and below.

Survival proportions: Survival of shCont shCD47-1

4. Some data were not representative. For example, in Fig. 1A, the tubulin in samples from Grade III and Grade IV tumors showed a dramatic difference, which may lead to the difference of THBS-1 levels between the tumors. In addition, whereas Fig 1A showed Grade III has more THBS-1 than Grade II tumors, Fig.1B showed the opposite staining.

Reply: Instead of showing western-blot, we analysed THBS1 expression in more patient samples from high grade and low-grade gliomas that include oligodendrogliomas and astrocytomas. These results show that THBS1 expression is significantly increased in grade IV gliomas. Accordingly we have removed the western-blots. The results were inserted in Figure 1A of the new manuscript and below.

5. When introducing the reasons for the investigation of the link between THBS-1 and TGF-beta, the authors may want to make a clear statement of the observed association between the two factors in the IRE-1 knock-down cells.

Reply: Our data show that THBS-1 is highly expressed in IRE1 knock-down cells compared to their control. This is consistent with our previous publication in PNAS (Auf et al, 2010) where the transcriptomic analysis revealed THBS-1 as the highest expressed gene in the IRE1dn cells. Furthermore, TGFβ1 is highly activated in these cells as shown in supplemental Figure 4B. THBS1 has been described TGFβ1 in several cell models (Schultz-Cherry, S., Lawler, J. & Murphy-Ullrich, J. E. *The type 1 repeats of thrombospondin 1 activate latent transforming growth factor-beta. The Journal of biological chemistry* 269, 26783-26788 (1994)). We therefore investigated the possibility that THBS-1 may also activate TGFβ in our GBM cells. However, our results unambiguously demonstrate that THBS1 silencing did not influence TGFβ activation as measured by Phospho-Smad2 nuclear translocation. We have stated this point more clearly in the manuscript (Discussion section, lines 305-311).

6. Given that the authors described the upregulated transcription of THBS-1 by TGF-beta, they should show some qPCR data to demonstrate the upregulated mRNA levels in core vs. invasive tumor areas.

Reply: We have again performed a new laser-capture microdissection of central and invasive areas in P3 tumours to respond to the criticism of the reviewer. Samples from invasive and central (core) areas were analysed for THBS1 expression by qPCR. The results are depicted in the figure below. The results clearly demonstrate that THBS1 expression is significantly increased in the invasive area in comparison to the tumour core. The results are presented in Figure 4B of the new manuscript and below.

7. In Page 9, Paragraph 1, the Fig 6C should be 5C, and the Fig 6D should be 5D.

Reply: This has been corrected.

Reviewer #2 (Remarks to the Author):

The data presented in the manuscript by Daubon et al. are not sufficiently supporting the conclusions of the authors, which compromises the significance of the manuscript.

Major comments:

The presentation of data is incomplete or insufficient at many places, for instance in the legend to Figure 1 it is claimed that n=12 different patient samples were analyzed for each grade. It is unclear why only two WB are shown, in addition the increase of THBS1 is accompanied by increases in tubulin when comparing Grade II with Grade IV. The IHC images are blurry and differences do not become obvious in Figure 1B. TMA of larger patient numbers need to be properly quantified. Similar comments apply at many other datasets when either data are missing or not supporting the statements made by the authors.

Several of the previous concerns remain unanswered, for instance the Gain-Of-Function experiment and the combinatory treatment experiment are still missing. It is not sufficient to state that experiments are ongoing.

Reply: We are providing now a great number of additional experiments that are also included to the response to reviewer 1. These experiments include:

- 1/ In vitro and in vivo experiments on CD47 knock-down cells: We have now provided additional results to reinforce the contention that tumour cell bound-CD47 is involved in THBS-1-mediated effects related to tumor cell invasion. We have carried out an invasion assay using P3 cells that were transduced with shRNA control or shRNA CD47 lentiviral vectors and done the assay in presence or absence of purified thrombospondin-1 protein. The results clearly indicate that knock-down of CD47 impairs THBS1-induced tumor cell invasion. We furthermore provided important additional in vivo results related to role of tumour-cell bound CD47 in glioblastoma development. P3 patient-derived CD47 knock-down tumour cells and control tumours cells were intracranially implanted in Ragγ2C-/- mice, and tumour development was followed over 100 days. As shown in figure 6D, survival was significantly increased in mice implanted with CD47 knock-down cells in comparison to control.

- 2/ Analysis of patient samples by IHC: Instead of showing western-blot, we analysed THBS1 expression in more patient samples from high grade and low-grade gliomas that include oligodendrogliomas and astrocytomas. These results show that THBS1 expression is significantly increased in grade IV gliomas (details of OA and AA). Accordingly, we have removed the western-blot. The results were inserted in Figure 1A and below.

- 3/ New Laser-capture Microdissection: We have again performed a new laser-capture microdissection of central and invasive areas in P3 tumours to respond to the criticism of the reviewer. Samples from invasive and central (core) areas were analysed for THBS1 expression by qPCR. The results are depicted in the figure below. The results clearly demonstrate that THBS1 expression is significantly increased (3-fold) in the invasive area in comparison to the tumour core.

- 4/ the Gain-Of-Function experiment: We have also performed a Gain-Of-Function experiment by expression THBS-1 using lentivirus. Cells were then investigated in *in vitro* invasion assay. As shown in Figure below and Figure 4B in the manuscript, invasion is significantly increased in THBS-1 overexpressing P3 cells.

In addition, we have treated THBS1 overexpressing cells with TAX2 peptide that inhibits the interaction with CD47. The results are shown in the figure below. TAX2 peptide clearly inhibits invasion in this assay. We did not include these results for clarity reasons.

- 5/ **the combinatory treatment experiment:** We have now completed the in vivo experiment using the combinatory treatment. The results are shown below and in Figure XX of the manuscript. Survival was significantly increased upon combinatory treatment (shTHBS1 and Avastin) in comparison to control. This new result was inserted in Figure 4F of the new manuscript and below.

Reviewers' Comments:

Reviewer #1:

Remarks to the Author:

The authors have addressed my concerns. The manuscript has been improved.

Reviewer #2:

Remarks to the Author:

My comments were not adequately addressed by the authors. For instance patient numbers in Figure 1 are still very low. In addition, the survival curves do not look different in shTHBS1-1 vs. shTHBS-1-1 + Bev. The statistical test applied is not adequate, log rank test needs to be performed when survival curves are compared.

Reply to the reviewers Comments:

Reviewer #1 (Remarks to the Author):

The authors have addressed my concerns. The manuscript has been improved.

Reply: we would like to thank the reviewer for his/her positive comments, which will allow acceptance of the manuscript.

Reviewer #2 (Remarks to the Author):

My comments were not adequately addressed by the authors. For instance patient numbers in Figure 1 are still very low. In addition, the survival curves do not look different in shTHBS1-1 vs. shTHBS1-1 + Bev. The statistical test applied is not adequate, log rank test needs to be performed when survival curves are compared.

Reply: the reviewer in his/her comment raises two concerns. First, the reviewer indicated that the patient numbers were too low in Figure 1. We therefore increased the patient number to 151 (57 of grade II, 27 from grade III and 37 from GBM). The graph is now included in the revised Figure 1.

Second, we have reanalyzed the statistics concerning the P3 shTHBS1-1 vs. P3 shTHBS1-1 + Bev survival curves. We show in both Log-Rank test and in the Gehan-Breslow-Wilcoxon test in below 0,005. We modified the figures accordingly (new Figures 4, 5 and 6).

We hope that we have now responded to all the concerns of the reviewer.